# Review of Chosen Isogeny-Based Cryptographic Schemes

Bartosz Drzazga *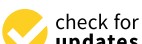 and Łukasz Krzywiecki 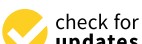

Faculty of Information and Communication Technology, Wrocław University of Science and Technology, 27 Wybrzeze Wyspianskiego St., 50-370 Wroclaw, Poland; lukasz.krzywiecki@pwr.edu.pl
* Correspondence: bartosz.drzazga@pwr.edu.pl

**Abstract:** Public-key cryptography provides security for digital systems and communication. Traditional cryptographic solutions are constantly improved, e.g., to suppress brute-force attacks. However, Shor's algorithm suited for quantum computers can break the bedrock of most currently used systems, i.e., the RSA problem and discrete logarithm problem. Post-quantum cryptography can withstand attacks carried out by quantum computers. Several families of post-quantum systems exist; one of them is isogeny-based cryptography. As a main contribution, in this paper, we provide a survey of chosen, fundamental isogeny-based schemes. The target audience of this review is researchers interested in practical aspects of this field of cryptography; therefore the survey contains exemplary implementations. Our goal was not to develop an efficient implementation, but to provide materials that make it easier to analyze isogeny-based cryptography.

**Keywords:** cryptography; post-quantum; isogenies; supersingular isogeny diffie–Hellman; SIDH



## 1. Introduction

Without secure solutions offered by asymmetric cryptography, our digital lives and digital businesses would be much less advanced than today. Public-key cryptography nowadays is essentially indispensable and used in almost all devices connected to a network. Public-key algorithms are a foundation of, e.g., secure and private web browsing over Transport Layer Security (TLS), encrypted messaging with OpenPGP, or any online service such as e-government. It is critically important to maintain sufficient security levels; thus, a lot of attempts have been made. As computational power and cost-efficiency of computers improve over time, key sizes are increased to suppress brute-force attacks. However, there are threats to classical asymmetric cryptography security that cannot be held back by a simple key size increase.

Public-key (asymmetric) cryptography is a system that utilizes a pair of keys. One of the keys may be known to everyone (public key), while the other key must not be known to anyone else but the owner (private key). Asymmetric cryptography is based on mathematical problems known as one-way functions, i.e., a function which is easy to compute on any input, but it is hard to compute its input given a result. Usually, to generate such a key pair, the private key is chosen at random, then the public key is computed as the result of the selected one-way function with the private key as the input. The most popular asymmetric cryptography systems are based on two hard problems: the RSA problem and discrete logarithm problem (DLP).

### 1.1. Problem Statement

In 1994, a groundbreaking algorithm suited for quantum computers was presented. Shor's algorithm [1] is a polynomial-time quantum computer algorithm for integer factorization. It means that cryptosystems such as RSA [2] could be broken by constructing a sufficiently large quantum computer.

Shor's algorithm is composed of two parts. The first part may be implemented on a classical computer and is responsible for turning the factorization problem into the problem

of finding the period of a function. The second part is a quantum subroutine that finds the period using the quantum Fourier transform. The period finding algorithm can be also used to break discrete logarithm problem instances by reduction.

### 1.2. Post-Quantum Alternatives

Post-quantum cryptography refers to cryptographic algorithms believed to be secure against attacks by a quantum computer. Initiatives such as NIST Post-Quantum Cryptography Competition [3] are an effort to standardize quantum-resistant schemes. In the third round out of this competition, six scheme families are still considered: lattice-based, multivariate, code-based, hash-based, zero-knowledge proofs, and isogeny-based. The focus of this article is isogeny-based cryptography.

One of the most promising types of post-quantum algorithms is lattice-based schemes (in the construction or in its security proof). Lattice-based cryptosystems include NTRU [4] for public-key encryption (PKE) and Falcon [5] for signatures.

Multivariate cryptography is a type of quantum-safe algorithms based on multivariate polynomials. This includes cryptographic schemes which are based on the difficulty of solving systems of multivariate equations. An example of this family is Rainbow [6] signature scheme.

Code-based cryptography involves cryptographic schemes based on error-correcting codes. Classic McEliece [7] encryption schemes belong to this type.

Hash-based cryptography includes cryptographic systems based on hash functions in contrast to number-theoretic schemes. SPHINCS+ [8] signature scheme is a representative of that family.

Zero-knowledge proof systems are based on zero-knowledge proofs and symmetric-key primitives such as hash functions and block ciphers. Picnic [9] signature scheme is of this type.

### 1.3. Contribution

The goal of this article is to introduce to the reader chosen constructions based on supersingular isogeny Diffie–Hellman (SIDH) [10] and SIDH itself. Our intention is to familiarize the reader with fundamental schemes based on isogenies: supersingular isogeny Diffie–Hellman key exchange (SIDH) [10], isogeny-based digital signature (IBDS) [11], strong designated verifier signature (SDVS) [12], undeniable signatures [13], and supersingular isogeny oblivious transfer (SIOT) [14]. Moreover, our goal is to illustrate high-level algorithms with implementations. We assume that such detailed level will help the reader to fully understand isogeny nuances and allow us to design fresh schemes with new functionalities. In this article:

- We discuss several fundamental solutions based on supersingular isogenies, provide summary of high-level mathematical aspects, constructions and algorithms.
- Most of the source publications use similar notations but there are some differences. We provide unified notation that allows easier comprehension and comparison of the solutions.
- We provide implementations of the chosen constructions in concise but clear code developed with easy to understand technologies. We discuss the solutions from the perspective of a programmer and comment the algorithms in more detail.

We note that implementations described in this article are not meant to assess the performance of the chosen schemes. Instead, we want to use code snippets as another tool (next to pseudocode) to describe the reviewed constructions. Pseudocode may not be clear enough in some cases, especially when complicated mathematical operations are involved. Instead, the reader can use our implementations to analyze the algorithms. The source codes can be executed and tweaked. We hope these implementations can be a helpful resource for every isogeny-based cryptography beginner or even for more experienced researchers. Some of the publications focused on isogeny-based cryptography provide low-level, fast implementations. These works are certainly a great feat; however,

the implementations in C, C++, and assembly are much harder to comprehend. We also provide time measurements of our implementations. These results are only for illustrative purposes and allow us to compare the chosen schemes. We do not intend to improve current state-of-the-art isogeny computations methods.

### 1.4. Organization of the Article

The article is organised into following sections. Section 2 contains characterisation of supersingular elliptic curve isogeny, mathematical definitions, isogeny algebra, and definitions of hard problems which provide security for isogeny-based cryptography. It also describes technologies used in implementations and common building blocks for all implemented solutions. Security assumptions for isogeny-based cryptography are stated in Section 3. In Section 4, SIDH, IBDS, SDVS, undeniable signatures, and SIOT schemes are presented. For each of the constructions, first a high-level description of the solution and algorithms are given. Then, implementation of the scheme is shown together with general comments and mappings to specific algorithms. Generic constructions used in the chosen schemes are presented in Section 5 together with an example of vulnerable application. Section 6 contains measurements of execution times of implemented cryptographic schemes. Finally, Section 7 contains conclusions.

## 2. Supersingular Elliptic Curve Isogeny Cryptography

Supersingular elliptic curve isogeny cryptography is a family of schemes based on the properties of supersingular elliptic curves and supersingular isogeny graphs. Isogeny-based schemes use the mathematics of supersingular elliptic curves to create a Diffie–Hellman-like key exchange. This means isogeny cryptography provides a straightforward quantum computing resistant replacement for widely used key exchange methods. Isogeny-based schemes boast significantly smaller key sizes than most of the popular post-quantum alternatives. There are two distinct families of systems based on isogeny algebra: supersingular isogeny Diffie–Hellman (SIDH) [10] and commutative supersingular isogeny Diffie–Hellman (CSIDH) [15]. The only scheme of this type still in NIST Post-Quantum Cryptography Competition is Supersingular Isogeny Key Encapsulation (SIKE) [16] which is based on SIDH. Many more cryptographic schemes similar to SIDH were proposed in the literature. We will focus only on the SIDH family.

Apart from public-key cryptography based on isogenies, other isogeny-based cryptographic schemes have been proposed. A recent example is Oblivious Pseudorandom Functions from Isogenies [17].

### 2.1. Definitions and Isogeny Algebra

This section contains essential mathematical Definitions related to isogenies. Further details on the mathematical foundations of isogenies are presented in, e.g., [10,18]. We use the following notation:

- Variables named $n, d, e, \ell, f$ represent integers, variables $p, q$ are prime numbers.
- Let $\phi$ be an isogeny.
- Let $y_1, \ldots, y_n \xleftarrow{R} Y$ denote each $y_i$ is sampled uniformly at random from the set $Y$.
- Elliptic curves are named $E$, the $j$-invariant of the elliptic curve $E$ is denoted as $j(E)$.
- $P, Q$ denote generators of a torsion subgroup.

The group of points on an elliptic curve $E$ over a finite field $\mathbb{F}_q$ of cardinality $q$ with a specified point $\mathcal{O}$ is denoted as $E(\mathbb{F}_q)$. The group contains a point at infinity and a set of points $(x, y)$ that satisfy the short Weierstrass form:

$$E/\mathbb{F}_q : y^2 = x^3 + ax + b, \tag{1}$$

where $a, b, x, y \in \mathbb{F}_q$. The *j*-invariant of an elliptic curve given by the Weierstrass equation is given by the formula:

$$j(E) = 1728 \frac{4a^3}{4a^3 + 27b^2}. \tag{2}$$

Two elliptic curves $E_1$ and $E_2$ are isomorphic to each other if and only if they have the same *j*-invariant, i.e., $j(E_1) = j(E_2)$.

Isogeny-based cryptography does not use elliptic curve's abelian group over point addition. Instead, it uses a map between curves called isogenies. An *isogeny* $\phi : E_1 \to E_2$ over $\mathbb{F}_q$ is a non-constant rational map from $E_1$ to $E_2$, which is a group homomorphism with a finite kernel. It follows that $\phi(\mathcal{O}_{E_1}) = \mathcal{O}_{E_2}$, where $\mathcal{O}$ denotes the identity element on an elliptic curve. An isogeny $\phi$ can be written as:

$$\phi = \left( \frac{f_1(x,y)}{g_1(x,y)}, \frac{f_2(x,y)}{g_2(x,y)} \right), \tag{3}$$

where $f_1, f_2, g_1, g_2$ are polynomials in variables $x, y$ with coefficients in $\mathbb{F}_q$. The degree of $\phi$ is its degree as an algebraic map, $deg(\phi) = max\{deg(f_1), deg(f_2)\}$. If the isogeny is separable, the degree of the isogeny is equal to the cardinality of its kernel. In this work, only separable isogenies are considered. An isogeny of degree $\ell$ is referred to as an $\ell$-isogeny. Two elliptic curves are $\ell$-isogenous if there exists an $\ell$-isogeny between them. Each $\ell$-isogeny $\phi : E_1 \to E_2$ has a dual isogeny $\hat{\phi} : E_2 \to E_1$ such that $\phi \circ \hat{\phi} = \hat{\phi} \circ \phi = [\ell]$ where $[\ell]$ is the multiplication by $\ell$ map. Isogeny $\hat{\phi}$ is also an $\ell$-isogeny.

For any natural $\ell$ we define $\ell$-torsion group $E[\ell]$ as the kernel of the multiplication by $\ell$ map over the algebraic closure $\bar{\mathbb{F}}_q$ of $\mathbb{F}_q$, i.e.,

$$E[\ell] = \{P \in E(\bar{\mathbb{F}}_q) : \ell P = \mathcal{O}\}. \tag{4}$$

The endomorphism ring $End(E)$ under the operations of point-wise addition and functional composition is defined as the set of all isogenies from $E$ to itself, defined over the algebraic closure $\bar{\mathbb{F}}_q$ of $\mathbb{F}_q$. If $dim(End(E)) = 4$ then $E$ is supersingular otherwise $E$ is ordinary. Two isogenous curves are always both ordinary or both supersingular.

The kernel $K$ of $\phi$ uniquely defines $\phi$ up to isomorphism so the codomain $E_2$ of the isogeny $\phi : E_1 \to E_2$ can be denoted as $E_1/K$. Any generator of the kernel will produce a unique isogeny up to isomorphism.

An isogeny graph is a graph in which nodes are *j*-invariants (representing isomorphism classes of elliptic curves), and in which edges are isogenies between them. The isogeny graph is undirected since each isogeny has a dual isogeny. It is hard to find a path of a given length between two random nodes in the isogeny graph. This hardness is the basis of isogeny-based cryptosystems.

### 2.2. Implementation Outline

For each of the chosen implemented schemes we include listings of code, corresponding comments, and mappings to algorithms. For each scheme, an exemplary execution is presented with one-run time measurements for order of magnitude reference, these were run on a single core of Intel Core i7-9750H processor. The following implementations are designated to be proof-of-concept implementations that help to understand the ideas behind the constructions and do not represent industrial grade level of security. Their main goal is to allow easy, fast modifications and experiments; thus, they use interpreted programming language and prioritize readability over execution speed and security. The implementations are not hardened against, e.g., timing attacks.

### 2.2.1. Common Building Blocks

All of the chosen schemes are implemented similarly, each implementation uses Python in version 3.9.5 and SageMath version 9.3. The implemented schemes are SIDH-based;

thus, common code related to parameters of SIDH is presented in this subsection. When not stated otherwise, implementations will use these shared parameters and functions.

For better performance, a different set of programming languages and libraries can be used. Microsoft's SIDH Library is a fast and portable library written in C that implements supersingular isogeny cryptographic schemes. Cloudflare provides SIDH library written in Go that ports portions of Microsoft's library. Those libraries make use of arithmetic written in assembly and allow us to compile the source code for a specific platform. Compared to SageMath, Microsoft's and Cloudflare's libraries are much better suited for a production-ready implementation; however, implementations written in Python and SageMath are easier to understand and experiment with.

### 2.2.2. SIDH Public Parameters

Public parameters that are common for most of the schemes described in Section 4 are a prime $p = \ell_A^{e_A} \ell_B^{e_B} \cdot f \pm 1$, a field $\mathbb{F}_{p^2}$, a supersingular elliptic curve $E(\mathbb{F}_{p^2})$, and $P_A, Q_A, P_B, Q_B$. This section contains listings with their definitions.

**Comment** (Listing 1): The above code creates a prime $p = \ell_A^{e_A} \ell_B^{e_B} \cdot f \pm 1$ using SIKEp434 [16] parameters, i.e., $p = 2^{216} 3^{137} - 1$. Then, a finite field Fp is created together with its quadratic extension Fp2, i.e., $\mathbb{F}_{p^2} = \mathbb{F}_p(i)$ with $i^2 + 1 = 0$. Finally, a starting curve E over that quadratic extension is created, defined by $y^2 = x^3 + x$. Integer $p$ must be prime and the starting elliptic curve must be supersingular, both these properties are tested in the assertions.

**Listing 1.** Public parameters.

```
1   f = 1
    lA = 2
    lB = 3
    eA = 216
5   eB = 137

    p = f * lA**eA * lB ** eB - 1
    assert p.is_prime()

10  Fp = GF(p)
    Fp2 = GF(p ** 2, 'i', modulus=x**2 + 1)

    E = EllipticCurve(Fp2, [1, 0])

15  assert E.is_supersingular()
```

**Comment** (Listing 2): Generators $(P_A, Q_A)$, $(P_B, Q_B)$ of $E[\ell_A^{e_A}]$ and $E[\ell_B^{e_B}]$, respectively, are generated dynamically with functions defined above. Function get_rand_point_ord returns a random point of order order on an elliptic curve E by first selecting a random point and then decreasing its order to the target value. This approach succeeds with high probability on the first try. Function get_random_base finds two points, $P$ and $Q$, that are independent of each other. For that, the Weil pairing is computed and its order must be equal to the order of points. Generators for Alice and Bob are computed in the following way.

**Listing 2.** Basis points functions.

```
1   def get_rand_point_ord(order, E, ord_oth):
        P = E.random_point()
        P_prime = ord_oth ** 2 * P
        while P_prime.order() != order:
5           P = E.random_point()
            P_prime = ord_oth ** 2 * P
        return P_prime

10  def get_random_base(order, E, ord_oth):
        P = get_rand_point_ord(order, E, ord_oth)
        Q = get_rand_point_ord(order, E, ord_oth)
        while P.weil_pairing(Q, order).multiplicative_order() != order:
            Q = get_rand_point_ord(order, E, ord_oth)
15      return P, Q
```

**Comment** (Listing 3): The above code introduces a Python dictionary params and a function get_other that allow us to conveniently store and access parameters of Alice (dictionary key A) and Bob (dictionary key B).

**Listing 3.** Public basis points.

```
1    PA, QA = get_random_base(lA**eA, E, f * lB ** eB)
     PB, QB = get_random_base(lB**eB, E, f * lA ** eA)

     params = {}
5    params['A'] = [PA, QA, lA, eA]
     params['B'] = [PB, QB, lB, eB]

     def get_other(name):
10       if name == 'A':
             return params['B']
         elif name == 'B':
             return params['A']
```

### 2.2.3. Isogeny Computation

SageMath allows to compute an isogeny using a library function, but that approach is inefficient for large parameters.

**Comment** (Listing 4): A better approach to computing $\ell^e$-isogeny is to compute a composition of $e$ individual $\ell$-isogenies. The function isogeny_graph_walk computes an isogeny, which is defined by a point P, from a curve E to a curve E_prime. It can also move points P_oth and Q_oth through that isogeny. The loop in line 5 in the above listing computes $e$ individual $\ell$-isogenies. The function returns a tuple matching SIDH public key, i.e., the final curve E_prime and the images of other party's public basis points.

**Listing 4.** Isogeny function.

```
1    def isogeny_graph_walk(E, P, l, e, P_oth = None, Q_oth = None):
         E_prime = E
         P_prime = P

5        for i in range(e):
             R = l ** (e - (i + 1)) * P_prime
             phi = E_prime.isogeny(R)
             P_prime = phi(P_prime)
             # assert P_prime.order() == l ** (e - (i + 1))
10           if (P_oth != None and Q_oth != None):
                 P_oth = phi(P_oth)
                 Q_oth = phi(Q_oth)
             E_prime = phi.codomain()

15       return (E_prime, P_oth, Q_oth)
```

## 3. Security of the Chosen Schemes

In this article, we do not repeat security proofs of the schemes we survey. We advise the readers interested in detailed security analysis of the schemes to refer to the corresponding original publications for security analysis and proofs.

The security of SIDH, IBDS, SDVS, undeniable signatures, and SIOT is provable and has been proven by reduction in which an adversary breaking each of the schemes could be used to break assumptions for isogeny-based cryptography. The following problems (from [10]) are believed to be intractable even for quantum computers and are the basis of security of isogeny-based cryptosystems.

**Definition 1** (Decisional Supersingular Isogeny (DSSI) problem)**.** *Let* $E, E_A$ *be supersingular curves defined over* $\mathbb{F}_{p^2}$*. Decide whether* $E_A$ *is* $\ell_A^{e_A}$*-isogenous to E.*

**Definition 2** (Computational Supersingular Isogeny (CSSI) problem)**.** *Let* $\phi_A : E \to E_A$ *be an isogeny whose kernel is* $\langle S_A \rangle$*, where* $S_A$ *is a random point with order* $\ell_A^{e_A}$*. Given* $E_A, \phi_A(P_B), \phi_A(Q_B)$*, find a generator of* $\langle S_A \rangle$*.*

**Definition 3** (Supersingular Computational Diffie–Hellman (SSCDH) problem). *Let $\phi_A : E \to E_A$ be an isogeny whose kernel is $\langle S_A \rangle$, where $S_A$ is a random point with order $\ell_A^{e_A}$, let $\phi_B : E \to E_B$ be an isogeny whose kernel is $\langle S_B \rangle$, where $S_B$ is a random point with order $\ell_B^{e_B}$. Given the curves $E_A, E_B$ and the points $\phi_A(P_B), \phi_A(Q_B), \phi_B(P_A), \phi_B(Q_A)$ find the j-invariant of $E / \langle S_A, S_B \rangle$.*

**Definition 4** (Supersingular Decision Diffie–Hellman (SSDDH) problem). *Given a tuple sampled with probability $1/2$ from one of the following two distributions, determine from which distribution the tuple is sampled:*

- *$(E_A, E_B, \phi_A(P_B), \phi_A(Q_B), \phi_B(P_A), \phi_B(Q_A), E_{AB})$, where $E_A, E_B, \phi_A(P_B), \phi_A(Q_B)$, $\phi_B(P_A), \phi_B(Q_A)$ are as in the SSCDH problem and $E_{AB} = E / \langle S_A, S_B \rangle$,*
- *$(E_A, E_B, \phi_A(P_B), \phi_A(Q_B), \phi_B(P_A), \phi_B(Q_A), E_C)$, where $E_A, E_B, \phi_A(P_B), \phi_A(Q_B)$, $\phi_B(P_A), \phi_B(Q_A)$ are as in the SSCDH problem and $E_C = E / \langle S_A', S_B' \rangle$, where $S_A'$ and $S_B'$ are random points with order $\ell_A^{e_A}$ and $\ell_B^{e_B}$, respectively.*

**Definition 5** (Decisional Supersingular Product (DSSP) problem). *Let $\phi : E \to E_3$ be an isogeny of degree $\ell_A^{e_A}$. Given $(E_1, E_2, \phi')$ sampled with probability $1/2$ from one of the following distributions, determine which distribution it is from.*

- *A random point $R$ of order $\ell_B^{e_B}$ is chosen and $E_1 = E / \langle R \rangle$, $E_2 = E_3 / \langle \phi(R) \rangle$, and $\phi' : E_1 \to E_2$ is an isogeny of degree $\ell_A^{e_A}$.*
- *$E_1$ is chosen randomly among curves of the same cardinality as $E$, and $\phi' : E_1 \to E_2$ is a random isogeny of degree $\ell_A^{e_A}$.*

On top of that, the undeniable signatures from [13] also use modified assumptions related to supersingular isogenies. Additionally, the schemes based on SIDH include security proofs of modifications and extensions that provide new functionalities.

The security of the chosen isogeny-based schemes is based on the problem of searching for an isogeny between elliptic curves having all information that a passive adversary could gather listening to the protocols' executions. To break these cryptosystems, one needs to find the secret isogeny between the starting elliptic curve $E$ and the elliptic curve in the public key, e.g., $E_A$. The adversary knows not only the preimage and image curves of the secret isogeny but also the public parameters, i.e., basis points and their images in public keys. One of the main assumption in the chosen schemes is that the additional information does not give any advantage to a passive adversary in solving the problem of computing the isogeny. This assumption remains valid; no known passive attacks use auxiliary points.

The authors of [19] show that an active adversary can reconstruct the secret if parties reuse their secret keys for many protocol executions. A malicious party can send to an honest party a public key with a modified auxiliary point and learn the secret key bit by bit. Currently, there is no known method for an honest party to check if the other public key is malicious. It is required that all parties use ephemeral secret keys or that a generic transformation is used, which allows one party to reuse their secret. SIKE applies such a transformation to SIDH.

The following approaches can be used for finding the secret isogeny.

- Brute-force attack: finding a path from the starting curve $E$ until reaching the curve in the public key, e.g., $E_A$. In brute-force attacks the number of isogeny computations is estimated to be linear to the size of the isogeny graph. It is not the best method for solving the isogeny problem because it does not take advantage of the fact that the secret isogeny has fixed and known degree. The task is in fact easier than a general problem of finding a path among all nodes. The parties take only $e_A$ or $e_B$ steps in an isogeny graph walk which is much shorter than the diameter of the graph.
- Meet in the middle: to depict the method let us use SIDH as an example, and fix $p = 2^{216} 3^{137} - 1$. We are searching for Alice's secret isogeny $\phi_A : E \to E_A$ of degree $2^{216}$. Instead of trying to find one path, two walks of 108 steps are performed starting from both $E$ and $E_A$. At some point, these walks meet in the middle, and with high

probability, these connected two paths form the walk that Alice took. To implement this attack one builds a table of all $2^{108}$-isogenous elliptic curves to $E_A$. Then, each $2^{108}$-isogeny is computed from $E$ until a match is found in the table. If $p = \ell_A^{e_A} \ell_B^{e_B} \cdot f \pm 1$ with $\ell_A^{e_A} \approx \ell_B^{e_B}$ is used, then $\ell_A^{e_A/2} \approx p^{1/4}$ so the attack runs in $O(p^{1/4})$ time and requires $O(p^{1/4})$ memory. The complexity remains the same if the other party is targeted.

- The authors of [20] argue that the generic meet in the middle algorithm also is not the best attack. They provide an nalysis showing that the van Oorschot-Wiener golden collision search is the best attack for finding the secret isogeny.

The hypothesis that these classical approaches cannot be improved using quantum computers is based on the following reasoning shown in [21]. To compute a walk of $e$ isogenies it is necessary to compute each isogeny successively. These computations cannot be parallelized because in each step the *j*-invariant changes. It also applies to quantum computers. Indeed, the author of [22] compares the best classical and quantum attacks and shows that "quantum computers don't really help".

## 4. Chosen Schemes

This article reviews by far the most popular isogeny-based protocol, i.e., SIDH. SIDH attracted almost the entire focus of isogeny-based cryptography in recent years. We also review a family of schemes based on SIDH. The main principle that determined which protocol is included in the review was the similarity to SIDH, e.g., in the form of public keys, and the flow of exchanged messages. In each of the reviewed protocols, it is clear how they build on top of SIDH. For this reason, e.g., commutative SIDH (CSIDH) was not included in the review since it differs greatly from SIDH, i.e., there are no auxiliary points exchanged, and it is possible to verify if the other party's key was generated honestly.

Since IBDS, SDVS, undeniable signatures, and SIOT are very much based on SIDH, we shall start the review with that scheme. It is the least complicated and we use it to also show how isogeny operations work in general.

### 4.1. Supersingular Isogeny Diffie–Hellman Key Exchange (SIDH)

#### 4.1.1. Construction and Algorithms

The Diffie–Hellman key exchange [23] is one of the earliest, practical, cryptographic key exchange protocols. It is also the first publicly known work that proposed the idea of a private key and a corresponding public key. In the traditional Diffie–Hellman, exponents commute to produce a shared secret $g^{ab} = (g^a)^b = (g^b)^a$ where $g$ is a generator of a multiplicative group $\mathbb{G}$ of prime order $q$ such that discrete logarithm problem and computational Diffie–Hellman problem hold, and $a, b$ are private keys of two parties. An efficient algorithm to solve the DLP would make DH key exchange insecure; thus, quantum-secure alternatives are needed.

The scheme that started an increased interest in isogeny based cryptography is supersingular isogeny Diffie–Hellman (SIDH) protocol [10] due to Jao and De Feo. More isogeny systems are based on SIDH so the initial setting is very often the same. SIDH works in the quadratic extensions of large prime fields $\mathbb{F}_p$, typically $\mathbb{F}_{p^2} = \mathbb{F}_p(i)$ with $i^2 + 1 = 0$; thus, elements can be represented as $a + bi$ where $a, b \in \mathbb{F}_p$. SIDH takes primes of the form

$$p = \ell_A^{e_A} \ell_B^{e_B} \cdot f \pm 1, \tag{5}$$

where $\ell_A, \ell_B$ are small primes (commonly 2 and 3) with $\ell_A^{e_A} \approx \ell_B^{e_B}$ and a small cofactor $f$ to ensure $p$ is prime (usually $f = 1$). The public parameters are a prime $p$, a supersingular curve $E(\mathbb{F}_{p^2})$ of order $(\ell_A^{e_A} \ell_B^{e_B} \cdot f)^2$, generators $P_A, Q_A$ of the $\ell_A^{e_A}$-torsion subgroup $E[\ell_A^{e_A}]$, and generators $P_B, Q_B$ of the $\ell_B^{e_B}$-torsion subgroup $E[\ell_B^{e_B}]$.

SIDH is similar to the classical Diffie–Hellman protocol. Two parties, named Alice and Bob, together compute a shared secret key. Alice creates her private key by choosing $m_A, n_A \xleftarrow{R} \mathbb{Z}/\ell_A^{e_A}\mathbb{Z}$, not both divisible by $\ell_A$, and computes an isogeny $\phi_A : E \to E_A$ with

kernel $\langle m_A P_A + n_A Q_A \rangle$. With almost no loss in generality, private keys with $m = 1$ can be used [24]. That convention is used throughout this work; thus, Alice selects $\mathsf{sk}_A \in \{0, 1, \dots, \ell_A^{e_A} - 1\}$ and computes

$$S_A = P_A + \mathsf{sk}_A\, Q_A. \tag{6}$$

Then, she computes her secret isogeny $\phi_A : E \to E_A$, where $E_A = E/\langle S_A \rangle$. This can be done as a composition of $e_A$ isogenies of degree $\ell_A$ (taking $e_A$ steps defined by $S_A$ in $\ell_A$-isogeny graph). Alice's public key is the image curve $E_A$ and the images of Bob's public basis points:

$$\mathsf{pk}_A = (E_A, P_B', Q_B') = (E_A, \phi_A(P_B), \phi_A(Q_B)). \tag{7}$$

Bob creates his private key by choosing $\mathsf{sk}_B \in \{0, 1, \dots, \ell_B^{e_B} - 1\}$ and computing

$$S_B = P_B + \mathsf{sk}_B\, Q_B. \tag{8}$$

Then, he computes his secret isogeny $\phi_B : E \to E_B$, where $E_B = E/\langle S_B \rangle$. This can be done as a composition of $e_B$ isogenies of degree $\ell_B$ (taking $e_B$ steps defined by $S_B$ in $\ell_B$-isogeny graph). Bob's public key is the image curve $E_B$ and the images of Alice's public basis points:

$$\mathsf{pk}_B = (E_B, P_A', Q_A') = (E_B, \phi_B(P_A), \phi_B(Q_A)). \tag{9}$$

Alice, having her secret integer $\mathsf{sk}_A$ and Bob's public key, computes the secret subgroup $S_A' = P_A' + \mathsf{sk}_A\, Q_A'$ on $E_B$. Then, she computes another secret isogeny $\phi_A' : E_B \to E_{AB}$, $E_{AB} = E_B/\langle S_A' \rangle$. The shared secret is $j_{AB} = j(E_{AB})$.

Bob computes the secret subgroup $S_B' = P_B' + \mathsf{sk}_B\, Q_B'$ on $E_A$, then computes secret isogeny $\phi_B' : E_A \to E_{BA}$, $E_{BA} = E_A/\langle S_B' \rangle$. The shared secret is $j_{BA} = j(E_{BA})$.

Computations carried out by Alice and Bob are an isogeny graph walk. Both parties start from the initial curve $E$, then Alice and Bob use their secret values and move to $E_A$ with $\phi_A$ and $E_B$ with $\phi_B$, respectively. After exchanging their public keys, they can continue their walks. Alice starts from $E_B$ and moves to $E_{AB}$ with $\phi_A'$, Bob proceeds mutatis mutandis and arrives in $E_{BA}$. Both parties end up in the same node of the isogeny graph so they can use $j(E_{AB}) = j(E_{BA})$ as their shared secret value. The isogeny graph walk of Alice and Bob is shown in Figure 1. A short description of SIDH is given in Table 1.

$$
\begin{array}{ccc}
E & \xrightarrow{\;\phi_A\;} & E/\langle S_A \rangle \\
\phi_B \downarrow & & \downarrow \phi_B' \\
E/\langle S_B \rangle & \xrightarrow{\;\phi_A'\;} & E/\langle S_A, S_B \rangle
\end{array}
$$

**Figure 1.** Isogeny graph walk of Alice and Bob.

**Remark 1.** *Both parties need to include images of each other's basis points through their secret isogenies. A composition of isogenies $\phi_A$ and $\phi_B$ does not make sense because of domain and codomain mismatch. Alice needs to start the second part of shared key computations from $E_B$, and Bob needs to start from $E_A$. This problem is solved by moving each other's basis points through the secret isogeny during key generation. In the end, they arrive at the same j-invariant. Except for that one difference, SIDH and classical DH share many common properties, like: both parties compute shared value, having a public key it is infeasible to compute the corresponding secret key, and knowing the secret key it is easy to compute the shared value.*

**Remark 2.** *Since SIDH and classical DH are so similar it seems natural we should be able to construct a lot of isogeny-based schemes based on SIDH just like in classical cryptography. DH is a foundation for many more constructions with a broad set of functionalities and security properties. However, this is not the case in isogeny-based cryptography. In classical cryptography we can use two operations, namely: exponentiation and addition of exponents. So far there is no second*

*operation in isogeny-based cryptography. It seems to be the main difficulty in developing new isogeny-based schemes.*

**Table 1.** SIDH scheme.

| Setup: |
|---|
| prime $p = \ell_A^{e_A} \ell_B^{e_B} \cdot f \pm 1$ |
| supersingular curve $E(\mathbb{F}_{p^2})$ |
| generators $P_A, Q_A$ of $E[\ell_A^{e_A}]$ |
| generators $P_B, Q_B$ of $E[\ell_B^{e_B}]$ |

| Key generation: | |
|---|---|
| Alice | Bob |
| 1. $\mathsf{sk}_A \xleftarrow{R} \mathbb{Z}/\ell_A^{e_A}\mathbb{Z}$ | 1. $\mathsf{sk}_B \xleftarrow{R} \mathbb{Z}/\ell_B^{e_B}\mathbb{Z}$ |
| 2. $S_A = P_A + \mathsf{sk}_A \, Q_A$ | 2. $S_B = P_B + \mathsf{sk}_B \, Q_B$ |
| 3. $\phi_A : E \to E_A = E/\langle S_A \rangle$ | 3. $\phi_B : E \to E_B = E/\langle S_B \rangle$ |
| 4. $\mathsf{pk}_A = (E_A, \phi_A(P_B), \phi_A(Q_B))$ | 4. $\mathsf{pk}_B = (E_B, \phi_B(P_A), \phi_B(Q_A))$ |

$$\xrightarrow{\mathsf{pk}_A}$$
$$\xleftarrow{\mathsf{pk}_B}$$

| Secret generation: | |
|---|---|
| 1. $S'_A = \phi_B(P_A) + \mathsf{sk}_A \, \phi_B(Q_A)$ | 1. $S'_B = \phi_A(P_B) + \mathsf{sk}_B \, \phi_A(Q_B)$ |
| 2. $\phi'_A : E_B \to E_{AB} = E/\langle S'_A \rangle$ | 2. $\phi'_B : E_A \to E_{BA} = E/\langle S'_B \rangle$ |
| 3. $j_{AB} = j(E_{AB})$ | 3. $j_{BA} = j(E_{BA})$ |

### 4.1.2. Implementation

SIDH is an interactive protocol between two parties, namely Alice and Bob. They both execute similar operations but using different parameters. Each party is implemented as an object of class Entity; however, objects representing Alice and Bob are created with different sets of parameters.

**Comment** (Listing 5): In the above listing, the object constructor __init__ corresponds to the key generation algorithm in Table 1. In line 9, a secret key sk is generated. In line 10, a generator S of a secret isogeny kernel is created. In the end, a call to isogeny_graph_walk with the secret key and other party's basis points produce the public key.

**Listing 5.** Class definition.

```
1   class Entity:
        def __init__(self, name):
            self.name = name
            self.P = params[name][0]
5           self.Q = params[name][1]
            self.l = params[name][2]
            self.e = params[name][3]

            self.sk = random.randrange(self.l ** self.e)
10          self.S = self.P + self.sk * self.Q
            # assert self.l ** self.e == self.S.order()
            self.pk = self.gen_pub_key(get_other(self.name))

15      def gen_pub_key(self, other):
            return isogeny_graph_walk(E, self.S, self.l, self.e, other[0], other[1])

        def gen_shared_key(self, peer):
20          S = peer.pk[1] + self.sk * peer.pk[2]
            shared_curve, _, _ = isogeny_graph_walk(peer.pk[0], S, self.l, self.e)
            return shared_curve.j_invariant()
```

A shared secret value is computed in the gen_shared_key method, which implements secret generation algorithm in Table 1. In line 20, a new generator S of a secret isogeny kernel is created. That generator is a point on the other party's public key curve. With that value, it is possible to compute the next isogeny in an isogeny graph. The target curves of both parties are in the same node of the graph; thus, its *j*-invariant is returned as a secret.

**Comment** (Listing 6): The above listing shows an example code that executes SIDH. First, an Entity A is created for Alice, then an Entity B for Bob. Alice can generate her shared value with Bob by passing object B as an argument to the gen_shared_key method in line 10. Bob computes his shared key with Alice in line 13 by passing object A as an argument to the gen_shared_key method. Both Alice and Bob must compute the same value, that condition is asserted in line 16.

**Listing 6.** SIDH key agreement.

```python
1   t0 = time.perf_counter()

    print('Started generation of PKA')
    A = Entity('A')
5
    print('Started generation of PKB')
    B = Entity('B')

    print('Started generation of secA')
10  secA = A.gen_shared_key(B)

    print('Started generation of secB')
    secB = B.gen_shared_key(A)

15  t1 = time.perf_counter()
    assert secA == secB
    print("Time elapsed (s):", t1 - t0)
```

**Comment** (Listing 7): The output of the above program shows that the assertion succeeded.

**Listing 7.** SIDH execution.

```
1   Started generation of PKA
    Started generation of PKB
    Started generation of secA
    Started generation of secB
5   Time elapsed (s): 7.953178946001572
```

*4.2. Isogeny-Based Digital Signature (IBDS)*

4.2.1. Construction and Algorithms

The scheme published in [11] by Yoo, Azarderakhsh, Jalali, Jao, and Soukharev is the first general-purpose digital signature scheme secure against quantum adversaries based on supersingular elliptic curve isogenies. The scheme uses Unruh's construction [25] of non-interactive zero-knowledge (NIZK) proofs applied to an interactive zero-knowledge proof presented in the same paper as SIDH [10]. This is similar to applying the Fiat–Shamir transform to an interactive zero-knowledge proof to build a secure signature scheme in classical cryptography. The Fiat–Shamir transform is only known to be secure against classical adversaries, as refs. [26,27] show this construction is not secure against quantum computers. The Unruh's construction is secure against quantum adversaries.

The public parameters of the isogeny-based digital signature (IBDS) are similar to the parameters of SIDH with one difference: only generators $P_B$ and $Q_B$ must be known publicly. IBDS simulates interactions in a sigma protocol using hashes. Parties run the protocol $2\lambda$ times, where $\lambda$ is the security parameter and the challenge domain is $\{0, 1\}$. Let $G, H$ be quantum random oracles, $G$ has the same domain and range, while $H$ outputs $2\lambda$ bits for challenges. A classical random oracle is modeled as a random hash function $O : \{0, 1\}^* \to \{0, 1\}^*$, it is possible to learn a value $O(x)$ by querying the classical state $x$. A quantum(-accessible) random oracle can be evaluated in superposition by submitting a quantum state. An attacker can query a random oracle with a superposition of many states, and the oracle must be evaluated at all points in the superposition. Unlike in Fiat–Shamir transform, $H$ will not be evaluated only on the commitments. The parameters for the hash function also include hashes from $G$ of the responses to each possible challenge for each commitment. The signature consists of the commitments, all possible challenges, hashed responses, and responses to the challenges given by $H$. The verifier gets the same

challenge bits from $H$ and verifies the responses in each round. The scheme consists of three algorithms: KeyGen, Sign, and Verify.

To implement KeyGen algorithm, shown in Algorithm 1, it suffices to use an implementation of Alice's key generation procedure from the SIDH protocol.

To generate keys, sample a random point $S$ of order $\ell_A^{e_A}$, compute the isogeny $\phi : E \to E/\langle S \rangle$ and calculate images of public points $P_B$ and $Q_B$. Sampling $S$ can be implemented as $S = P_A + \mathsf{sk}\, Q_A$ for some $\mathsf{sk} \in \mathbb{Z}/\ell_A^{e_A}\mathbb{Z}$. The public key is composed of the target curve $E/\langle S \rangle$ and public generators $P_B, Q_B$ moved through $\phi$, i.e., $\phi(P_B), \phi(Q_B)$.

The signing procedure uses Unruh's construction and emulates the execution of an interactive zero-knowledge proof of knowledge.

For each of $2\lambda$ rounds of the sigma protocol, a signer chooses a random point $R$ of order $\ell_B^{e_B}$ (this corresponds to Bob's key generation from SIDH) and continues with all computations in the sigma protocol. Then, the signer hashes all the responses. In the last step, data from all rounds is hashed with the message $m$ to obtain the challenge bits $J_1 || \ldots || J_{2\lambda}$. All of the operations until the very last hash function call can be precomputed in parallel even before the message is known.

---

**Algorithm 1:** KeyGen()

---

Sample $\mathsf{sk} \xleftarrow{R} \mathbb{Z}/\ell_A^{e_A}\mathbb{Z}$
Compute $S = P_A + \mathsf{sk}\, Q_A$
Compute the isogeny $\phi : E \to E/\langle S \rangle$
$\mathsf{pk} \leftarrow (E/\langle S \rangle, \phi(P_B), \phi(Q_B))$
**return** $(\mathsf{pk}, \mathsf{sk})$

---

Each round of the signing procedure is an independent isogeny graph walk. Figure 2 shows a walk computed in one round of the simulation. The upper part in the isogeny graph is fixed during signing, i.e., the starting point $E$, the private isogeny $\phi$, and a part of the signer's public key $E/\langle S \rangle$. The lower part of the figure is random and changes in each round. This simulates the interactions of Alice with different parties.

$$
\begin{array}{ccc}
E & \xrightarrow{\ \phi\ } & E/\langle S \rangle \\
\psi \downarrow & & \downarrow \psi' \\
E/\langle R \rangle & \xrightarrow{\ \phi'\ } & E/\langle R, S \rangle
\end{array}
$$

**Figure 2.** Isogeny graph walk in one round of the signing algorithm.

The signature verification corresponds to the verification of data (commitments, challenges, and responses) in each round of the simulated sigma protocol.

First, a verifier computes the same hash value $J_1 || \cdots || J_{2\lambda}$ and checks each ZKP round based on data in the signature. The verification starts with checking the challenge bit for this round, then the hash value for the response selected by the challenge bit is verified. Depending on the challenge bit, properties of the corresponding response are tested. If all checks succeed, then the signature is correct.

### 4.2.2. Implementation

Before implementations of algorithms are given, it is important to define scheme-specific functions used in those algorithms.

**Comment** (Listing 8): In the listing, functions $G$ and $H$ correspond to the quantum random oracles $G, H$ of the scheme, respectively.

**Listing 8.** Random oracles functions.

```
1   def G(x):
        h_obj = SHA3_256.new()
        h_obj.update(f'{x}'.encode())
        return h_obj.hexdigest()

    def H(x, len):
        h_obj = SHA3_256.new()
        h_obj.update(x.encode())
10      h = h_obj.digest()
        bytes_as_bits = ''.join(format(byte, '08b') for byte in h)
        return [int(c) for c in bytes_as_bits][:len]
```

Both functions are based on a SHA3-256 hash function output. In case of function G, the result is just a SHA3-256 output. This does not influence a proof-of-concept implementation, although random oracle *G* has the same domain and range according to the scheme. That property of *G* is used only in the proof that this construction is secure in the quantum oracle model. The proof exploits the fact that the random oracle *G* is indistinguishable from a random permutation, and replaces *G* with an efficiently invertible function, which is unnoticeable by any quantum PPT adversary. That modification allows the hashes to be inverted to obtain the hidden responses in the adversary's forged proof. Function H returns an array of integers in $\{0, 1\}$ of length len, i.e., $2\lambda$. These values correspond to the challenge bits $J_1||\ldots||J_{2\lambda}$. Usage of SHA3-256 in function H sets the limit of security parameter to $\lambda = 128$. The exact definitions of functions G and H can be easily modified to achieve higher security levels.

**Comment** (Listing 9): Helper functions used in the verification procedure are also defined. Function generates_kernel checks if a point P generates a kernel of the $\ell^e$-isogeny from *E*1 to *E*2 by checking if *E*2 is the same node as $E1/\langle P \rangle$ in an isogeny graph. Despite the fact that SageMath provides a method to calculate a point's order, it is too expensive to use it in the testing if the point is of the correct order. Instead, function has_order is defined to test if a point P has order $\ell^e$ by checking if *e* multiplications by scalar *l* bring the point to a point at infinity.

**Listing 9.** Property testing functions.

```
1   def generates_kernel(P, E1, E2, l, e):
        E2_prime, _, _ = isogeny_graph_walk(E1, P, l, e)
        return E2.j_invariant() == E2_prime.j_invariant()

    def has_order(P, l, e):
        for _ in range(e):
            if P.is_zero():
                return False
10          P *= l
        if P.is_zero():
            return True
        return False
```

A signer role is implemented similarly to the class Entity from Listing 5, a secret and a public keys are generated with exactly the same code. A scheme specific method gen_shared_key is replaced with a sign method.

**Comment** (Listing 10): The above method implements Algorithm 2 and signs a message m with a signer's secret key self.sk. The loop in line 8 executes $2\lambda$ simulations, in each round a new random point R is selected. Then, an isogeny graph walk is computed together with isogenies, auxiliary points, and elliptic curves. Starting from line 18, all values are packed into structures for commitments, challenges, and corresponding responses. At the end of each round, in line 25, every response is hashed. In line 26, challenge bits J are computed. A signature consists of commitments, challenges, and responses' hashes computed in simulations, together with responses selected by the challenge bits.

**Listing 10.** Definition of signing method.

```python
def sign(self, m):
    other = get_other(self.name)
    P_oth, Q_oth, l_oth, e_oth = other[0], other[1], other[2], other[3]
    com = [0] * 2*lamb
    ch = [0] * 2*lamb
    resp = [0] * 2*lamb
    h = [0] * 2*lamb
    for i in range(2*lamb):
        r = random.randrange(l_oth**e_oth)
        R = P_oth + r*Q_oth

        ER, psi_S1, psi_S2 = isogeny_graph_walk(E, R, l_oth, e_oth, self.P, self.Q)
        psi_S = psi_S1 + self.sk*psi_S2

        phi_R = self.pk[1] + r*self.pk[2]
        ERS, _, _ = isogeny_graph_walk(self.pk[0], phi_R, l_oth, e_oth)

        com[i] = (ER, ERS)
        c0 = random.randint(0, 1)
        c1 = 1 - c0
        ch[i] = (c0, c1)
        resp[i] = ((R, phi_R), psi_S)
        if ch[i][0] == 1:
            resp[i] = (resp[i][1], resp[i][0])
        h[i] = (G(resp[i][0]), G(resp[i][1]))
    J = H(f'{self.pk}{m}{com}{ch}{h}', 2*lamb)

    ret_resp = []
    for i in range(2*lamb):
        ret_resp.append(resp[i][J[i]])
    return (com, ch, h, ret_resp)
```

---

**Algorithm 2:** $\mathsf{Sign}(\mathsf{sk}, m)$

> **for** $i = 1$ *to* $2\lambda$ **do**
>> Pick a random point $R$ of order $\ell_B^{e_B}$ (sample $r \xleftarrow{R} \mathbb{Z}/\ell_B^{e_B}\mathbb{Z}$, then $R = P_B + rQ_B$)
>> Compute the isogeny $\psi : E \to E/\langle R \rangle$ Compute either $\phi' : E/\langle R \rangle \to E/\langle R, S \rangle$
>>　or $\psi' : E/\langle S \rangle \to E/\langle R, S \rangle$
>> $(E_1, E_2) \leftarrow (E/\langle R \rangle, E/\langle R, S \rangle)$
>> $com_i \leftarrow (E_1, E_2)$
>> $ch_{i,0} \xleftarrow{R} \{0, 1\}$
>> $(resp_{i,0}, resp_{i,1}) \leftarrow ((R, \phi(R)), \psi(S))$
>> **if** $ch_{i,0} = 1$ **then**
>>> $swap(resp_{i,0}, resp_{i,1})$
>>
>> **end**
>> $h_{i,j} \leftarrow G(resp_{i,j})$
>
> **end**
> $J_1 || \ldots || J_{2\lambda} \leftarrow H(\mathsf{pk}, m, (com_i)_i, (ch_{i,j})_{i,j}, (h_{i,j})_{i,j})$
> **return** $\sigma \leftarrow ((com_i)_i, (ch_{i,j})_{i,j}, (h_{i,j})_{i,j}, (resp_{i,J_i})_i)$

---

**Comment** (Listing 11): The verification Algorithm 3 is implemented in a form of a standalone function as the verification procedure is universal. The verification of a signature sigma for a message m, allegedly signed by a signer signer, starts with unpacking the signature in line 4. With those values, together with the signer's public key, challenge bits J are computed in line 6. The loop in line 8 tests each round of simulation embedded into the signature. Depending on a challenge bit, the response is verified against the correct hash value in line 9. Then, the commitment and the response are unpacked. Next, the properties of each element are tested with helper functions. In case all the checks succeed, the signature is accepted as valid.

**Listing 11.** Verification function.

```python
def verify(signer, m, sigma):
    other = get_other(signer.name)
    l_oth, e_oth = other[2], other[3]
    com, ch, h, ret_resp = sigma

    J = H(f'{signer.pk}{m}{com}{ch}{h}', 2*lamb)

    for i in range(2*lamb):
        if h[i][J[i]] != G(ret_resp[i]):
            return False

        E1, E2 = com[i][0], com[i][1]
        if ch[i][J[i]] == 0:
            R, phi_R = ret_resp[i]
            if not has_order(R, l_oth, e_oth) or not has_order(phi_R ,l_oth, e_oth):
                return False
            if not generates_kernel(R, E, E1, l_oth, e_oth) or not generates_kernel(phi_R,
            ↪  signer.pk[0], E2, l_oth, e_oth):
                return False
        else:
            psi_S = ret_resp[i]
            if not has_order(psi_S, signer.l, signer.e):
                return False
            if not generates_kernel(psi_S, E1, E2, signer.l, signer.e):
                return False
    return True
```

**Comment** (Listing 12): The above listing shows an example code that executes the IBDS scheme. First, a signer A is created in line 4 using Alice's parameters from SIDH. The signer signs a message m by passing it to the sign method in line 10. Verification of a message m and its signature sigma over the signer's public key is executed in line 13. The program also tries to verify the signature for a modified message in line 18.

**Listing 12.** Test program of IBDS.

```python
t0 = time.perf_counter()

print('Started Keygen')
A = Entity('A')

lamb = 8
m = 'test'

print('Started Sign')
sigma = A.sign(m)

print('Started Verify')
res = verify(A, m, sigma)

t1 = time.perf_counter()
print("Time elapsed (s):", t1 - t0)
print('Original message', res)
print('Altered message', verify(A, m+m, sigma))
```

---

**Algorithm 3:** Verify$(\mathsf{pk}, m, \sigma)$

---

$J_1 || \ldots || J_{2\lambda} \leftarrow H(\mathsf{pk}, m, (com_i)_i, (ch_{i,j})_{i,j}, (h_{i,j})_{i,j})$

**for** $i = 1$ **to** $2\lambda$ **do**

    **check** $h_{i,J_i} = G(resp_{i,J_i})$

    **if** $ch_{i,J_i} = 0$ **then**

        Parse $(R, \phi(R)) \leftarrow resp_{i,J_i}$

        **check** $R, \phi(R)$ have order $\ell_B^{e_B}$

        **check** $R$ generates the kernel of the isogeny $E \to E_1$

        **check** $\phi(R)$ generates the kernel of the isogeny $E/\langle S \rangle \to E_2$

    **else**

        Parse $\psi(S) \leftarrow resp_{i,J_i}$

        **check** $\psi(S)$ have order $\ell_A^{e_A}$

        **check** $\psi(S)$ generates the kernel of the isogeny $E_1 \to E_2$

    **end**

**end**

**if** *all checks succeed* **then**

    **return** 1

**else**

    **return** 0

**end**

---

**Comment** (Listing 13): This listing shows that the correct signer-message-signature triple is valid, while modifications of the message are detected.

**Listing 13.** IBDS scheme execution.

```
1   Started Keygen
    Started Sign
    Started Verify
    Time elapsed (s): 110.57361834400001
5   Original message True
    Altered message False
```

*4.3. Strong Designated Verifier Signature (SDVS)*

4.3.1. Construction and Algorithms

The scheme proposed in [12] by Sun, Tian, and Wang is the first strong designated verifier signature scheme that may be secure against a quantum computer. The scheme combines two ideas, the first one is a method of constructing an SDVS based on Diffie–Hellman key exchange, and the second concept used in the proposed construction is SIDH instead of classical key exchange. The public parameters and assumptions of the SDVS scheme are almost the same as in SIDH. Let $p = \ell_A^{e_A} \ell_B^{e_B} \cdot f \pm 1, E(\mathbb{F}_{p^2}), P_A, Q_A, P_B, Q_B$ be defined exactly as in SIDH. Let $H : \{0,1\}^* \to \{0,1\}^k$ be a secure hash function, where $k$ is a security parameter.

The key generation Algorithm 4 executes similarly to the SIDH key generation. A signer follows the computations of Alice while a verifier computes as Bob.

The signing algorithm is based on the secret generation steps of Alice from SIDH in Table 1 and introduces operations based on a value shared between a signer and a designated verifier.

To sign a message $m$ for a designated verifier, a signer follows SIDH secret generation and obtains a $j$-invariant $j_{AB}$ of the final node in an isogeny graph walk of the signer with the designated verifier's public key. The designation part of the signature creation is achieved through the use of the designated verifier's public key in the computations. The signature is created as $\sigma = H(m||j_{AB})$, where $||$ denotes bits concatenation.

In this SDVS scheme, the signature is an HMAC of a message with a shared secret as the key, and the key is established in the execution of SIDH protocol. Both a signer and a designated verifier follow computations of Alice and Bob, respectively, and calculate

the shared *j*-invariant. This value is then concatenated with a message, and the final bits are hashed.

The verification algorithm is also based on the secret generation steps from SIDH but follows computations of Bob in Table 1.

A signature $\sigma$ for a message *m* can be verified by a designated verifier as follows: using the verifier's secret key and the signer's public key compute an isogeny graph walk similarly to Bob's computations in SIDH, compute a *j*-invariant $j_{BA}$ of the final curve, then $\sigma' = H(m||j_{BA})$. The signature is correct if and only if $\sigma = \sigma'$. The designated verifier can simulate a correct signature for *m* by outputting $\sigma'$ as the signature.

### 4.3.2. Implementation

The SDVS scheme is a simple extension of SIDH key exchange; thus, its implementation is heavily based on the SIDH implementation from Section 4.1.2. A signer reuses the code of Alice, while a designated verifier reuses the code of Bob. Both new methods for signing and verification use a call to a hash function *H*, defined as follows.

**Comment** (Listing 14): Function *H* is a simple wrapper for SHA3-256 hash function that returns hash output for a message *m* concatenated with a *j*-invariant j. Signing and verification algorithms are implemented as new methods of the class Entity from Listing 5, i.e., sign and verify, respectively.

**Listing 14.** Helper functions of SDVS.

```
1  def H(m, j):
       h_obj = SHA3_256.new()
       h_obj.update(f'{m}{j}'.encode())
       return h_obj.hexdigest()
```

**Comment** (Listing 15): The method sign implements Algorithm 5. It takes a message m and a designated verifier verifier. The function computes a shared secret value of the signer and the verifier according to SIDH. In line 3, a hash over the message and the shared value is computed, then the message-hash pair is returned. The verify method implements Algorithm 6. It takes a signer signer, a message m, and a signature sigma. The verifier computes a secret key shared with the signer and a new signature sigma_prime. If both hashes sigma and sigma_prime are equal, then the signature is correct.

**Listing 15.** Implementation of signing and verification in SDVS.

```
1  def sign(self, m, verifier):
       j = self.gen_shared_key(verifier)
       sigma = H(m, j)
       return m, sigma

5
   def verify(self, signer, m, sigma):
       j = self.gen_shared_key(signer)
       sigma_prime = H(m, j)
10     return sigma == sigma_prime
```

---

**Algorithm 4:** KeyGen()

| Signer | Designated verifier |
| --- | --- |
| $\mathsf{sk}_A \xleftarrow{R} \mathbb{Z}/\ell_A^{e_A}\mathbb{Z}$ | $\mathsf{sk}_B \xleftarrow{R} \mathbb{Z}/\ell_B^{e_B}\mathbb{Z}$ |
| $S_A = P_A + \mathsf{sk}_A\,Q_A$ | $S_B = P_B + \mathsf{sk}_B\,Q_B$ |
| $\phi_A : E \to E_A = E/\langle S_A \rangle$ | $\phi_B : E \to E_B = E/\langle S_B \rangle$ |
| $\mathsf{pk}_A = E_A, \phi_A(P_B), \phi_A(Q_B)$ | $\mathsf{pk}_B = E_B, \phi_B(P_A), \phi_B(Q_A)$ |

**return** $((\mathsf{pk}_A, \mathsf{sk}_A), (\mathsf{pk}_B, \mathsf{sk}_B))$

---

**Algorithm 5:** $\mathsf{Sign}(\mathsf{sk}_A, \mathsf{pk}_B, m)$

$S'_A = \phi_B(P_A) + \mathsf{sk}_A \, \phi_B(Q_A)$
$\phi'_A : E_B \to E_{AB} = E/\langle S'_A \rangle$
$j_{AB} = j(E_{AB})$
$\sigma = H(m || j_{AB})$
**return** $\sigma$

**Algorithm 6:** $\mathsf{Verify}(\mathsf{sk}_B, \mathsf{pk}_A, m, \sigma)$

$S'_B = \phi_A(P_B) + \mathsf{sk}_B \, \phi_A(Q_B)$
$\phi'_B : E_A \to E_{BA} = E/\langle S'_B \rangle$
$j_{BA} = j(E_{BA})$
$\sigma' = H(m || j_{BA})$
**return** $\sigma == \sigma'$

**Comment** (Listing 16): The above listing shows an example code that executes the SDVS scheme. A signer A is created in line 4 using the parameters of Alice. In line 7, a designated verifier is created with the parameters of Bob. The signer signs a message m for the designated verifier B in line 12. The verifier B checks the signature sig for the message m from the signer A in line 15. The designated verifier also tries to verify the signature for an altered message in line 22. In line 25, a verifier C tries to verify a correct message and signature, but for the verifier B.

**Listing 16.** Test program of SDVS.

```
1    t0 = time.perf_counter()

     print('Started Keygen A')
     A = Entity('A')
5
     print('Started Keygen B')
     B = Entity('B')

     m = "test message"
10
     print('Started signing')
     m, sig = A.sign(m, B)

     print('Started verification')
15   is_correct = B.verify(A, m, sig)

     print('Is signature correct?', is_correct)

     t1 = time.perf_counter()
20   print("Time elapsed (s):", t1 - t0)

     print('Is signature correct for a modified message?', B.verify(A, m+m, sig))

     C = Entity('B')
25   print('Verification by party C', C.verify(A, m, sig))
```

**Comment** (Listing 17): This listing shows the designated verifier can verify a correct signature and detect modifications in a message. Another party was unable to verify a correct message-signature pair.

**Listing 17.** SDVS scheme execution.

```
1    Started Keygen A
     Started Keygen B
     Started signing
     Started verification
5    Is signature correct? True
     Time elapsed (s): 7.820308166999894
     Is signature correct for a modified message? False
     Verification by party C False
```

### 4.4. Undeniable Signatures

4.4.1. Construction and Algorithms

The scheme presented in [13] by Jao, and Soukharev is an example of isogeny-based undeniable signature. It is related to SIDH in assumptions and basic principles but introduces more changes than previously described schemes. Let $p$ be a prime of the form

$$p = \ell_A^{e_A} \ell_M^{e_M} \ell_C^{e_C} \cdot f \pm 1, \tag{10}$$

fix a supersingular curve $E$ over $\mathbb{F}_{p^2}$, such that its order is divisible by $(\ell_A^{e_A} \ell_M^{e_M} \ell_C^{e_C})^2$, generators $\{P_A, Q_A\}$ of $E[\ell_A^{e_A}]$, $\{P_M, Q_M\}$ of $E[\ell_M^{e_M}]$, and $\{P_C, Q_C\}$ of $E[\ell_C^{e_C}]$, let $H : \{0,1\}^* \to \mathbb{Z}$. In general, the scheme is designed to use points in $\langle P_A, Q_A \rangle$ for keys, points in $\langle P_M, Q_M \rangle$ are associated with messages, and points in $\langle P_C, Q_C \rangle$ are linked to commitments.

A signer creates a key pair (Algorithm 7) following the steps of Alice's key generation in SIDH key exchange. Instead of using Bob's basis points, $\{P_C, Q_C\}$ are moved through the secret isogeny.

To sign a message $m$ (Algorithm 8), a signer computes a final node in an isogeny graph walk similarly to an isogeny graph in SIDH. The difference is that instead of Bob's side, values are based on a hash value over the message. The signature consists of $E_{AM}$ and auxiliary points $\phi_{M,AM}(\phi_M(P_C))$, $\phi_{M,AM}(\phi_M(Q_C))$. Isogeny graph walk of signing procedure is shown in Figure 3.

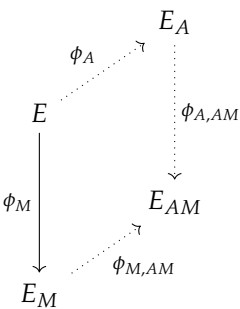

**Figure 3.** Isogeny graph walk during signature generation.

---

**Algorithm 7:** KeyGen()

---

Sample $\mathsf{sk}_A \xleftarrow{R} \mathbb{Z}/\ell_A^{e_A}\mathbb{Z}$
Compute $S_A = P_A + \mathsf{sk}_A\, Q_A$
Compute the isogeny $\phi_A : E \to E_A = E/\langle S_A \rangle$
$\mathsf{pk}_A \leftarrow (E/\langle S_A \rangle, \phi_A(P_C), \phi_A(Q_C))$
**return** $(\mathsf{pk}_A, \mathsf{sk}_A)$

---

**Algorithm 8:** Sign($S_A, m$)

---

$h = H(m)$
$S_M = P_M + hQ_M$
$\phi_M : E \to E_M = E/\langle S_M \rangle$
$\phi_{M,AM} : E_M \to E_{AM} = E/\langle \phi_M(S_A) \rangle$
$\phi_{A,AM} : E_A \to E_{AM} = E/\langle \phi_A(S_M) \rangle$
**return** $\sigma = (E_{AM}, \phi_{M,AM}(\phi_M(P_C)), \phi_{M,AM}(\phi_M(Q_C)))$

---

The signature verification is done in an interactive confirmation of $E_{AM}$. Isogenies used to produce $E_{AM}$ must not be revealed; thus, this curve will be blinded with a third step in an isogeny graph walk. A signer computes commitment as in Algorithm 9.

---

**Algorithm 9:** Confirmation-commitment($S_A, m, \sigma$)

---

$\mathsf{sk}_C \xleftarrow{R} \mathbb{Z}/\ell_C^{e_C}\mathbb{Z}$,
$S_C = P_C + \mathsf{sk}_C\, Q_C$,
$\phi_C : E \to E_C = E/\langle S_C \rangle$,
$\phi_{M,MC} : E_M \to E_{MC} = E_M/\langle \phi_M(S_C) \rangle = E_C/\langle \phi_C(S_M) \rangle$,
$\phi_{A,AC} : E_A \to E_{AC} = E_A/\langle \phi_A(S_C) \rangle = E_C/\langle \phi_C(S_A) \rangle$,
$\phi_{MC,AMC} : E_{MC} \to E_{AMC} = E_{AM}/\langle \phi_{C,MC}(\phi_C(S_A)) \rangle$,
**return** comm $= (E_C, E_{AC}, E_{MC}, E_{AMC}, ker(\phi_{C,MC}))$

---

A verifier chooses at random $b \xleftarrow{R} \{0, 1\}$. If $b = 0$, the signer returns resp $= ker(\phi_C)$, if $b = 1$, the signer returns resp $= ker(\phi_{C,AC})$ and the verifier can continue with verification shown in Algorithm 10.

---

**Algorithm 10:** Confirmation-verification(comm, $b$, resp, $m, \sigma$)

---

**if** *b = 0* **then**
    compute $\phi_{A,AC}$ using the signer's public key
    compute $\phi_{M,MC}$ using knowledge of $ker(\phi_M)$
    compute $\phi_{AM,AMC}$ using the auxiliary points in the signature
    **check** if each isogeny maps between the corresponding two curves in comm
    compute $\phi_{C,MC}$ using knowledge of $ker(\phi_C)$
    **check** if $\phi_{C,MC}$ matches kernel in comm
**end**
**if** *b = 1* **then**
    compute $\phi_{MC,AMC}$
    compute $\phi_{AC,AMC}$ using knowledge of $ker(\phi_M)$
    **check** if each of $\phi_{C,AC}, \phi_{MC,AMC}, \phi_{AC,AMC}$ maps between the corresponding
      two curves in comm
**end**
**if** *all checks succeed* **then**
    **return** 1
**end**

---

The isogeny graph walk computed during confirmation protocol is shown in Figure 4.

The disavowal protocol is executed when a signer wants to prove that a falsified signature $(E_F, F_P, F_Q)$ for a message $m$ is invalid. Here, $E_F, F_P, F_Q$ correspond to $E_{AM}$ and auxiliary points $\phi_{M,AM}(\phi_M(P_C))$, $\phi_{M,AM}(\phi_M(Q_C))$, respectively. The signer shows that $E_F$ is an invalid signature without revealing correct $E_{AM}$. Elliptic curve $E_{AM}$ must not be disclosed because it is a part of a valid signature over $m$ that the signer does not intend to sign. This can be achieved by blinding $E_{AM}$ once again to obtain $E_{AMC}$. Additional information allows the verifier to compute $E_{FC}$ and then verify that $E_{FC} \neq E_{AMC}$.

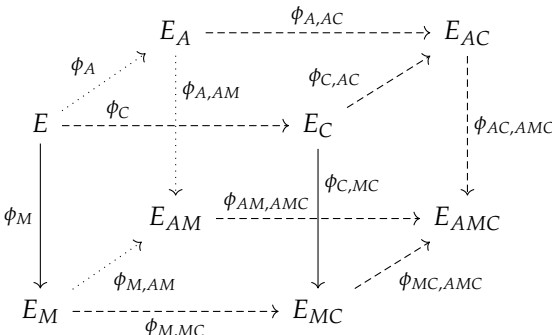

**Figure 4.** Confirmation protocol.

The disavowal protocol begins similarly to confirmation protocol, i.e., the signer sends $E_C, E_{AC}, E_{MC}, E_{AMC}, ker(\phi_{C,MC})$ as the commitment, more detailed steps are provided in Algorithm 9.

A verifier chooses at random $b \xleftarrow{R} \{0,1\}$. If $b = 0$, the signer returns resp $= ker(\phi_C)$, if $b = 1$, the signer returns resp $= ker(\phi_{C,AC})$ and the verifier checks all properties as given in Algorithm 11.

---

**Algorithm 11:** Disavowal-verification(comm, $b$, resp, $m$, $\sigma$)

**if** *b = 0* **then**
　compute $\phi_C$
　compute $\phi_{M,MC}$ using knowledge of $ker(\phi_M)$
　compute $\phi_{A,AC}$ using the signer's public key
　compute $\phi_F : E_F \rightarrow E_{FC} = E_F / \langle F_P + sk_C F_Q \rangle$
　**check** if each isogeny maps between the corresponding two curves in comm
　recompute $\phi_{C,MC}$ using knowledge of $ker(\phi_C)$
　**check** if $\phi_{C,MC}$ matches kernel in comm
　**check** that $E_{FC} \neq E_{AMC}$
**end**
**if** *b = 1* **then**
　compute $\phi_{MC,AMC}$
　compute $\phi_{AC,AMC}$ using knowledge of $ker(\phi_M)$
　**check** these isogenies map to $E_{AMC}$
**end**
**if** *all checks succeed* **then**
　**return** 1
**end**

---

The isogeny graph walk computed during disavowal protocol is shown in Figure 5.

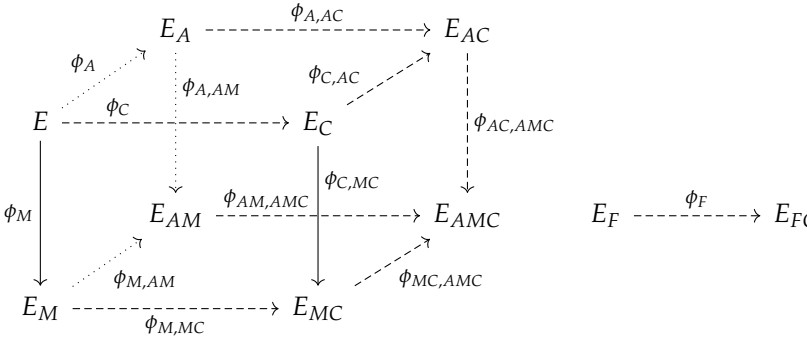

**Figure 5.** Disavowal protocol.

### 4.4.2. Implementation

The undeniable signature scheme needs the most changes among all schemes described in this work. The implementation of the scheme differs in many aspects compared to SIDH, including public parameters.

**Comment** (Listing 18): The code creates a prime $p = \ell_A^{e_A} \ell_M^{e_M} \ell_C^{e_C} \cdot f \pm 1$ of a suitable form for the scheme, this implementation uses toy-size parameters. The starting elliptic curve is created just like in SIDH scheme.

**Listing 18.** Public parameters.

```
1    f = 1
     lA = 3
     lM = 2
     lC = 5
5    eA = 3
     eM = 4
     eC = 2

     p = f * lA**eA * lM**eM * lC**eC - 1
10   assert p.is_prime()

     Fp = GF(p)
     Fp2 = GF(p ** 2, 'i', modulus=x**2 + 1)

15   E = EllipticCurve(Fp2, [1, 0])

     assert E.is_supersingular()
```

**Comment** (Listing 19): Public base points $(P_A, Q_A), (P_M, Q_M), (P_C, Q_C)$ can be computed dynamically with appropriate calls to get_random_base function from common code Section 2.2.1:

**Listing 19.** Public basis points.

```
1    PA, QA = get_random_base(lA**eA, E, f * lM**eM * lC**eC)
     PM, QM = get_random_base(lM**eM, E, f * lA**eA * lC**eC)
     PC, QC = get_random_base(lC**eC, E, f * lA**eA * lM**eM)
```

**Comment** (Listing 20): Functions and methods implementing algorithms of the scheme will use the following helper functions. Function H is a wrapper function for SHA3-256 which returns a hash as an integer value from a correct range. Function generates_kernel is used for the verification of $ker(\phi_{C,MC})$ in a commitment and works exactly the same as the function with identical name in Listing 9. Function isogeny_graph_walk is similar to the function with identical name in Section 2.2.3, a method of passing points to move through a new isogeny is different.

**Listing 20.** Helper functions.

```
1    def H(x):
         h_obj = SHA3_256.new()
         h_obj.update(f'{x}'.encode())
         return int.from_bytes(h_obj.digest(), "big") % lC**eC
5

     def generates_kernel(P, E1, E2, l, e):
         E2_prime, _ = isogeny_graph_walk(E1, P, l, e)
         return E2.j_invariant() == E2_prime.j_invariant()
10

     def isogeny_graph_walk(E, P, l, e, points=[]):
         E_prime = E
         P_prime = P
15
         for i in range(e):
             R = l ** (e - (i + 1)) * P_prime
             phi = E_prime.isogeny(R)
             P_prime = phi(P_prime)
20           # assert P_prime.order() == l ** (e - (i + 1))
             points = [phi(x) for x in points]
             E_prime = phi.codomain()

         return (E_prime, points)
```

In the implementation, isogenies are called explicitly only in the isogeny_graph_walk function. The rest of the code operates on elliptic curves, their *j*-invariants, and points on elliptic curves. This is because the schemes use $\ell^e$-isogenies, while such isogeny never exists in a running program. An $\ell^e$-isogeny is always computed as a composition of *e* individual $\ell$-isogenies. However, the undeniable signatures scheme needs to move a varying number of points through $\ell^e$-isogenies; thus, the isogeny_graph_walk function accepts a list of points.

All computations of a signer are grouped as methods of class Signer.

**Comment** (Listing 21): Object constructor __init__ creates a private key of a signer and computes a corresponding public key in the gen_pub_key method as a triple of the secret isogeny's target curve and images of $P_C, Q_C$.

**Listing 21.** Signer class.

```python
class Signer:
    def __init__(self):
        self.sk = random.randrange(lA ** eA)
        self.S = PA + self.sk * QA
        # assert lA ** eA == self.S.order()
        self.pk = self.gen_pub_key()

    def gen_pub_key(self):
        EA, points = isogeny_graph_walk(E, self.S, lA, eA, [PC, QC])
        return (EA, points[0], points[1])

    def sign(self, m):
        h = H(m)
        SM = PM + h*QM
        EM, phiM_points = isogeny_graph_walk(E, SM, lM, eM, [PA, QA, PC, QC])
        EAM, phiMAM_points = isogeny_graph_walk(EM, phiM_points[0] + self.sk*phiM_points[1], lA, eA,
        ↪ [*phiM_points[2:]])
        return (EAM, *phiMAM_points)

    def get_commitment_conf(self, m, signature):
        new_sig = self.sign(m)
        # it does not make sense to confirm a signature for m
        # in case the signer concludes they did not sign m
        if new_sig != signature:
            raise Exception("Signer: This is not my signature")
        return self.get_commitment_disavow(m, signature)

    def get_commitment_disavow(self, m, signature):
        h = H(m)

        sk_C = random.randrange(lC**eC)
        SC = PC + sk_C*QC

        EC, phiC_points = isogeny_graph_walk(E, SC, lC, eC, [PM, QM, self.S])
        phiC_SM = phiC_points[0] + h*phiC_points[1]
        EMC, phiCMC_points = isogeny_graph_walk(EC, phiC_SM, lM, eM, [phiC_points[2]])
        EAC, phiCAC_points = isogeny_graph_walk(EC, phiC_points[2], lA, eA, [phiC_SM])
        EAMC, _ = isogeny_graph_walk(EMC, phiCMC_points[0], lA, eA)

        self.response = [[sk_C, phiCAC_points[0]], phiC_points]

        return (EC, EAC, EMC, EAMC, phiC_SM)

    def get_response(self, b):
        return self.response[b]
```

The sign method implements Algorithm 8. A secret integer for a message m together with a generator SM are computed in lines 15 and 16. The generator is used to calculate an elliptic curve EM in line 17 and images of basis points. Then, the basis points connected with commitments are moved from the elliptic curve EM to EAM. These points and the elliptic curve EAM together are returned as a signature.

Both confirmation and disavowal protocols use commitments of the same form. The get_commitment_conf methods is used during confirmation protocol. Before a signer proceeds with its execution, a signature in question is checked. In case a signer concludes this is not a legitimate signature, an exception is raised in line 27 and no commitment is sent. When a signer decides to participate in the confirmation protocol, a commitment is returned according to the scheme. Algorithm 9 is implemented in the get_commitment_disavow method. In line 34, a random value is generated. Then, elliptic curves EC, EMC, EAC, and EAMC are computed. All auxiliary points that are needed are also moved through the corresponding isogenies. In line 43,

responses are saved for later use. Curves EC, EMC, EAC, EAMC, and kernel $ker(\phi_{C,MC})$ are returned as a commitment.

The get_response method simply returns a correct response saved before, depending on a challenge bit b.

The confirmation and disavowal protocols are implemented as methods of the class Verifier. However, both methods could be plain functions as the scheme does not authenticate a verifier. The above listing presents the confirmation protocol.

**Comment** (Listing 22): The confirmation method takes a message m, a signature signature, and a signer Signer as parameters. The verifier asks the signer to execute the confirmation protocol with a call to the get_commitment_conf method in line 4 with the message and the signature. In case the signer throws an exception, the execution is aborted. This corresponds to a situation in which the signer claims the signature is forged. In that scenario, both parties could engage in the disavowal protocol covered below. Otherwise, a commitment is returned and stored in comm. In lines 8 and 9, the verifier coins at random a challenge bit b and asks for a response resp. Starting from line 12, Algorithm 10 is implemented. Depending on the challenge bit b, the correct response is unpacked. Then, codomains of isogenies are computed according to the algorithm. Both loops in lines 24 and 41 verify recomputed elliptic curves against the commitment. In line 29, the kernel from the commitment is also verified.

**Listing 22.** Confirmation method.

```
1    def confirmation(self, m, signature, Signer):
        for _ in range(lamb):
            try:
                comm = Signer.get_commitment_conf(m, signature)
5           except Exception as e:
                print(e)
                return False
            b = random.randrange(2)
            resp = Signer.get_response(b)
10
            h = H(m)
            if b == 0:
                sk_C = resp[0]
                SC = PC + sk_C*QC
15
                SM = PM + h*QM
                EM, phiM_points = isogeny_graph_walk(E, SM, lM, eM, [SC])
                EC, phiC_points = isogeny_graph_walk(E, SC, lC, eC, [SM])

20              EAC, _ = isogeny_graph_walk(Signer.pk[0], Signer.pk[1] + sk_C*Signer.pk[2], lC, eC)
                EMC, _ = isogeny_graph_walk(EM, phiM_points[0], lC, eC)
                EAMC, _ = isogeny_graph_walk(signature[0], signature[1] + sk_C*signature[2], lC, eC)

                for curve_comm, curve_verify in zip(comm, [EC, EAC, EMC, EAMC]):
25                  if curve_comm.j_invariant() != curve_verify.j_invariant():
                        return False

                EMC2, _ = isogeny_graph_walk(EC, phiC_points[0], lM, eM)
                if EMC2.j_invariant() != EMC.j_invariant() or not generates_kernel(comm[4], EC, EMC2, lM,
                ↪  eM):
30                  return False
            else:
                phiC_PM, phiC_QM, ker_phiCAC = resp
                EAC, phiCAC_points = isogeny_graph_walk(comm[0], ker_phiCAC, lA, eA, [comm[4]])
                EMC, phiCMC_points = isogeny_graph_walk(comm[0], phiC_PM + h*phiC_QM, lM, eM,
                ↪  [ker_phiCAC])
35
                EAMC, _ = isogeny_graph_walk(EMC, phiCMC_points[0], lA, eA)
                EAMC2, _ = isogeny_graph_walk(comm[1], phiCAC_points[0], lM, eM)

                curves_comm = [comm[1], comm[2], comm[3], comm[3]]
40              curves_ver = [EAC, EMC, EAMC, EAMC2]
                for curve_comm, curve_verify in zip(curves_comm, curves_ver):
                    if curve_comm.j_invariant() != curve_verify.j_invariant():
                        return False
        return True
```

The disavowal protocol is implemented in the disavowal method defined below.

**Comment** (Listing 23): The disavowal method also takes a message m, a signature signature, and a signer Signer as parameters. The verifier asks the signer to execute the disavowal protocol with a call to the get_commitment_disavow method in line 3 with the message and the signature. This time no exception is expected at that point. In lines 4 and 5, the verifier coins at random a challenge bit b and asks for a response resp. Starting from line 8, Algorithm 11 is implemented. The verification in disavowal protocol executes somewhat similarly, i.e., in one case a different elliptic curve is computed and the verification is extended. In line 18, the elliptic curve $E_{FC}$ (see Figure 5) is computed, then it is verified against $E_{AMC}$ in line 31. The else clause in line 33, similarly to the confirmation protocol, checks that the right face of the cube commutes, i.e., the commitment is correct.

**Listing 23.** Disavowal method.

```
1   def disavowal(self, m, signature, Signer):
        for _ in range(lamb):
            comm = Signer.get_commitment_disavow(m, signature)
            b = random.randrange(2)
5           resp = Signer.get_response(b)

            h = H(m)
            if b == 0:
                sk_C = resp[0]
10              SC = PC + sk_C*QC

                SM = PM + h*QM
                EM, phiM_points = isogeny_graph_walk(E, SM, lM, eM, [SC])
                EC, phiC_points = isogeny_graph_walk(E, SC, lC, eC, [SM])
15
                EAC, _ = isogeny_graph_walk(Signer.pk[0], Signer.pk[1] + sk_C*Signer.pk[2], lC, eC)
                EMC, _ = isogeny_graph_walk(EM, phiM_points[0], lC, eC)
                EFC, _ = isogeny_graph_walk(signature[0], signature[1] + sk_C*signature[2], lC, eC)
                EMC2, _ = isogeny_graph_walk(EC, phiC_points[0], lM, eM)
20
                curves_comm = [comm[0], comm[1], comm[2], comm[2]] # EC, EAC, EMC, EMC
                curves_ver = [EC, EAC, EMC, EMC2]
                for curve_comm, curve_verify in zip(curves_comm, curves_ver):
                    if curve_comm.j_invariant() != curve_verify.j_invariant():
25                      return False

                if EMC2.j_invariant() != EMC.j_invariant() or not generates_kernel(comm[4], EC, EMC2, lM,
                ↪  eM):
                    return False

30              EAMC, _ = isogeny_graph_walk(EAC, resp[1], lC, eC)
                if EFC.j_invariant() == EAMC.j_invariant():
                    return False
            else:
                phiC_PM, phiC_QM, ker_phiCAC = resp
35              EAC, phiCAC_points = isogeny_graph_walk(comm[0], ker_phiCAC, lA, eA, [comm[4]])
                EMC, phiCMC_points = isogeny_graph_walk(comm[0], phiC_PM + h*phiC_QM, lM, eM,
                ↪  [ker_phiCAC])

                EAMC, _ = isogeny_graph_walk(EMC, phiCMC_points[0], lA, eA)
                EAMC2, _ = isogeny_graph_walk(comm[1], phiCAC_points[0], lM, eM)
40
                if EAMC.j_invariant() != comm[3].j_invariant() or EAMC2.j_invariant() !=
                ↪  comm[3].j_invariant():
                    return False
        return True
```

**Comment** (Listing 24): The above listing shows an example code executing the undeniable signatures scheme. In the first line, the security parameter lamb (or $\lambda$) that controls the number of rounds is set to 128. In lines 5 and 6, two signers S1 and S2 are created. The while loop in 7 ensures these two signers have different public keys. It is forced by the usage of such small parameters in the implementation. A verifier V is created in line 9, and both signers sign the same message m in lines 13 and 14. In line 17, the confirmation protocol is used with a valid message-signature-signer combination. The confirmation method calls from lines 18 and 19 test the cases where a signer is presented with a modified message and someone's else signature, respectively. In line 25, the disavowal method is called with a correct message-signature-signer combination. Lines 26 and 27 execute disavowal protocol for a modified message and different signature cases, respectively.

**Listing 24.** Test program of undeniable signatures.

```
1   lamb = 128
    t0 = time.perf_counter()

    print('Started Keygen')
5   S1 = Signer()
    S2 = Signer()
    while S1.pk[0].j_invariant() == S2.pk[0].j_invariant():
        S2 = Signer()
    V = Verifier()
10
    m = 'test'
    print('Started Sign')
    sig1 = S1.sign(m)
    sig2 = S2.sign(m)
15
    print('Started Confirmation')
    res1 = V.confirmation(m, sig1, S1)
    res2 = V.confirmation(m+m, sig1, S1)
    res3 = V.confirmation(m, sig2, S1)
20  print('Original message', res1)
    print('Altered message', res2)
    print('Other signature', res3)

    print('Started Disavowal')
25  res4 = V.disavowal(m, sig1, S1)
    res5 = V.disavowal(m+m, sig1, S1)
    res6 = V.disavowal(m, sig2, S1)
    print('Original message', res4)
    print('Altered message', res5)
30  print('Other signature', res6)

    t1 = time.perf_counter()
    print("Time elapsed (s):", t1 - t0)
```

**Comment** (Listing 25): This listing presents the output of the above program. It shows that the correct signature was successfully confirmed while an altered message and somebody's else signature were detected. The signer refused to proceed with the execution and threw exceptions. The signer was unable to prove that they did not sign a previously signed message. However, disavowal succeeded for an altered message and an invalid signature.

**Listing 25.** Undeniable signatures scheme execution.

```
1   Started Keygen
    Started Sign
    Started Confirmation
    Signer: This is not my signature
5   Signer: This is not my signature
    Original message True
    Altered message False
    Other signature False
    Started Disavowal
10  Original message False
    Altered message True
    Other signature True
    Time elapsed (s): 19.333842884999967
```

*4.5. Supersingular Isogeny Oblivious Transfer (SIOT)*

4.5.1. Construction and Algorithms

The scheme due to Barreto, Oliveira, and Benits proposed in [14] is Oblivious Transfer (OT) protocol based on supersingular isogenies. This construction is a combination of the OT scheme of Chou and Orlandi [28], and SIDH. Once again, the public parameters of SIOT are based on SIDH. In OT scheme the sender (Alice) has two messages $x_0, x_1$ and wants to send one of them to the receiver (Bob). Bob can choose which message he will receive and will never learn the other message. The choice of Bob is unknown to Alice. Let $\mathcal{M}$ be a set of all messages with binary strings of fixed length and $(x_0, x_1) \in \mathcal{M}$. Let $\mathcal{C}$ be a set of all ciphertexts with binary strings of fixed length and $(c_1, c_2) \in \mathcal{C}$. Alice and Bob agree to use a symmetric encryption scheme $Enc(j, x)$ taking a shared key $j$ and a message $x$ to encrypt. The shared key $j$ is a $j$-invariant of a shared supersingular elliptic curve.

The scheme needs a secure coin-flipping subprotocol so Alice and Bob can agree on an ephemeral, uniformly random bit string $w$. That part of the scheme is not covered in this article, the authors of SIOT suggest to use, e.g., Wagner's bit commitment protocol [29]. Alice and Bob also need to agree on a deterministic algorithm that maps $w$ into a pair of points $U, V \in E_B[\ell_A^{e_A}]$.

The key generation algorithm for Alice executes just like in SIDH. However, Bob on top of SIDH computations will also modify his key depending on his choice $b$ of the message to receive.

Alice, upon receiving Bob's public key, is able to generate the same points $U, V$. Now Alice needs to encrypt her two messages. To do so securely, she first computes two keys (*j*-invariants) by taking isogeny graph walks.

Bob can follow SIDH steps to compute the elliptic curve shared with Alice-but only the one that corresponds to his choice $b$.

### 4.5.2. Implementation

Although SIOT is heavily based on SIDH, it introduces a lot of modifications in the middle of computations. On top of that, computations of Alice and Bob are not symmetrical. Those properties make it difficult to reuse previous implementations for SIOT. Most of helper functions (e.g., isogeny walk) and global parameters of SIDH can be reused. The following listing presents new and modified helper functions for SIOT.

**Comment** (Listing 26): The get_key function is a wrapper for the SHA3-256 hash function that returns 16 byte long keys for any input j. The get_u_v function is used by Alice and Bob to obtain the shared points $U, V$. Both parties hash the bit string $w$ to compute the coefficients $\alpha$ and $\beta$. Bob computes $U = \alpha G_B + \beta H_B$ and $V = -\frac{\alpha}{\beta} V$. The coefficients $\alpha, \beta$ are computed as SHA3-256 hashes of $w$ with appended bitstring alpha in line 10 and beta in line 14, respectively. Alice can use exactly the same computations to obtain the correct points. She will use $\hat{G}_B$ and $\hat{H}_B$ instead of $G_B$ and $H_B$. It will work in both possible values of blinded public key of Bob. After obtaining coefficients, $U$ is computed in line 15, and $V$ in line 18. Note that $\alpha, \beta \in \mathbb{Z}/\ell_A^{e_A}\mathbb{Z}$, and inversion of $\beta$ is also calculated modulo $\ell_A^{e_A}$.

**Listing 26.** Helper function for SIOT.

```
1    def get_key(j):
         h_obj = SHA3_256.new()
         h_obj.update(f'{j}'.encode())
         return h_obj.digest()[:16]
5

     def get_u_v(w, Gb, Hb):
         h_obj = SHA3_256.new()
         h_obj.update(w + b"alpha")
10       alpha = int.from_bytes(h_obj.digest(), "big") % lA**eA

         h_obj = SHA3_256.new()
         h_obj.update(w + b"beta")
         beta = int.from_bytes(h_obj.digest(), "big") % lA**eA
15       U = alpha*Gb + beta*Hb

         beta_inv = inverse_mod(beta, lA**eA)
         V = - alpha * beta_inv * U
         return U, V
```

The following listing contains the modified definition of SIDH's Entity class which will be a base class for Alice and Bob.

**Comment** (Listing 27): The only difference between SIDH and SIOT in the definition of class Entity is that the method gen_shared_key takes an elliptic curve and two points instead of the other party's object. It is because Bob works on the public key of Alice, but Alice works on the blinded public key of Bob.

**Listing 27.** Entity class for SIOT.

```python
1   class Entity:
        def __init__(self, name):
            self.name = name
            self.P = params[name][0]
5           self.Q = params[name][1]
            self.l = params[name][2]
            self.e = params[name][3]

            self.sk = random.randrange(self.l ** self.e)
10          self.S = self.P + self.sk * self.Q
            # assert self.l ** self.e == self.S.order()
            self.pk = self.gen_pub_key(get_other(self.name))

15      def gen_pub_key(self, other):
            return isogeny_graph_walk(E, self.S, self.l, self.e, other[0], other[1])

        def gen_shared_key(self, E, P, Q):
20          S = P + self.sk * Q
            shared_curve, _, _ = isogeny_graph_walk(E, S, self.l, self.e)
            return shared_curve.j_invariant()
```

**Comment** (Listing 28): The class Alice inherits from the Entity class and is a container for computations of Alice. In this case an object representing Alice is initiated with a given bitstring w. In practice, Alice and Bob would use a secure coin-flipping protocol to agree on a uniformly random bit string. We did not implement it as it is not relevant to isogenies.

**Listing 28.** Alice class for SIOT.

```python
1   class Alice(Entity):
        def __init__(self, w):
            super().__init__('A')
            self.w = w
5           self.x = [os.urandom(16) for _ in range(2)]

        def produce_response(self, pk_hat_B):
            self.U, self.V = get_u_v(self.w, pk_hat_B[1], pk_hat_B[2])
10          c = []
            for i in [0, 1]:
                j = self.gen_shared_key(pk_hat_B[0], pk_hat_B[1] + i * self.U, pk_hat_B[2] + i * self.V)
                cipher = AES.new(get_key(j), AES.MODE_EAX)
                c.append({
15                  'ciphertext': cipher.encrypt_and_digest(self.x[i]),
                    'nonce': cipher.nonce
                })
            return c
```

In line 5, Alice creates two random messages $x$. Those messages are encrypted according to Algorithm 12 in the produce_response function that takes the blinded public key of Bob pk_hat_B. She computes two $j$-invariants in line 12 and encrypts the $i$th message in line 15. The key used for encryption is derived from the corresponding $j$-invariant in line 13.

---

**Algorithm 12:** Encrypt($\mathsf{sk}_A, x_0, x_1, \hat{\mathsf{pk}}_B$)

---

$U, V \xleftarrow{R} E_B[\ell_A^{e_A}]$
$\forall i \in \{0, 1\}$
$S'_{A_i} = (\hat{G}_B + iU) + \mathsf{sk}_A(\hat{H}_B + iV)$
$\phi'_{A_i} : E_B \rightarrow E_{BA_i} = E/\langle S'_{A_i} \rangle$
$j_i = j(E_{BA_i})$
$c_i = \mathsf{Enc}(j_i, x_i)$
**return** $c_0, c_1$

---

**Comment** (Listing 29): The class Bob inherits from the Entity class and is a container for computations of Bob. Since the focus is on isogeny-based cryptography, Bob is given the same $w$ just like Alice. In line 5, Bob chooses the message b and in lines 7–10 blinds his public key according to Algorithm 13. The function get_result is used to decrypt the chosen message according to Algorithm 14. The shared $j$-invariant is computed in line 14, which is then used to derive the correct AES key in line 15. The $b$th message is decrypted in line 16.

**Listing 29.** Bob class for SIOT.

```
1   class Bob(Entity):
        def __init__(self, w):
            super().__init__('B')
            self.w = w
5           self.b = random.randint(0, 1)

            self.U, self.V = get_u_v(self.w, self.pk[1], self.pk[2])
            self.G_hat_B = self.pk[1] - self.b * self.U
            self.H_hat_B = self.pk[2] - self.b * self.V
10          self.pk_hat = (self.pk[0], self.G_hat_B, self.H_hat_B)

        def get_result(self, c, pk_A):
            j = self.gen_shared_key(*pk_A)
15          cipher = AES.new(get_key(j), AES.MODE_EAX, nonce=c[self.b]['nonce'])
            return cipher.decrypt(c[self.b]['ciphertext'][0])
```

---

**Algorithm 13:** $\mathsf{KeyGen}()$

| Sender | Receiver |
|---|---|
| $x_0, x_1 \in \mathcal{M}$ | $b \in \{0, 1\}$ |
| $\mathsf{sk}_A \xleftarrow{R} \mathbb{Z}/\ell_A^{e_A}\mathbb{Z}$ | $\mathsf{sk}_B \xleftarrow{R} \mathbb{Z}/\ell_B^{e_B}\mathbb{Z}$ |
| $S_A = P_A + \mathsf{sk}_A\, Q_A$ | $S_B = P_B + \mathsf{sk}_B\, Q_B$ |
| $\phi_A : E \to E_A = E/\langle S_A \rangle$ | $\phi_B : E \to E_B = E/\langle S_B \rangle$ |
| $G_A = \phi_A(P_B), H_A = \phi_A(Q_B)$ | $G_B = \phi_B(P_A), H_B = \phi_B(Q_A)$ |
| $\mathsf{pk}_A = (E_A, G_A, H_A)$ | |
| | $U, V \xleftarrow{R} E_B[\ell_A^{e_A}]$ |
| | $\hat{G}_B = G_B - bU$ |
| | $\hat{H}_B = H_B - bV$ |
| | $\hat{\mathsf{pk}}_B = (E_B, \hat{G}_B, \hat{H}_B)$ |

**return** $\left( (\mathsf{pk}_A, \mathsf{sk}_A), (\hat{\mathsf{pk}}_B, \mathsf{sk}_B) \right)$

---

**Algorithm 14:** $\mathsf{Decrypt}(\mathsf{sk}_B, c_0, c_1, \mathsf{pk}_A)$

$S_B' = G_A + \mathsf{sk}_B\, H_A$
$\phi_B' : E_A \to E_{AB} = E/\langle S_B' \rangle$
$j_b = j(E_{AB})$
$x_b = \mathsf{Enc}^{-1}(j_b, c_b)$
**return** $c_0, c_1$

---

**Comment** (Listing 30): The listing shows an example code that executes SIOT. First, a bitstring $w$ is created. Note that in practice Alice and Bob would use a secure coin-flipping protocol to compute that value. Then, objects for Alice A and Bob B are created. In line 13, the method produce_response of Alice is used to compute ciphertexts c. The blinded key of Bob pk_hat is used as the input. The get_result method of Bob with public key of Alice is used in line 16 to decrypt the chosen message xb. Bob must obtain the same value as the one in possession of Alice. The assertion in line 19 checks that condition.

**Listing 30.** Test program of SIOT.

```
1    # Alice and Bob need to use a secure coin-flipping protocol to agree on a uniformly random bit string
     ↪  w that is unique for each session
     w = bytes.fromhex('ef60425e20f1493a51850443f0175acb')

     t0 = time.perf_counter()
5
     print('Started generation of PKA')
     A = Alice(w)

     print('Started generation of PK_hat_B')
10   B = Bob(w)

     print('Started generation of encrypted messages c')
     c = A.produce_response(B.pk_hat)

15   print('Started decryption of xb')
     xb = B.get_result(c, A.pk)

     t1 = time.perf_counter()
     assert xb == A.x[B.b]
20   print("Time elapsed (s):", t1 - t0)
```

**Comment** (Listing 31): The output of the above program is presented in the listing, it shows that the assertion succeeded.

**Listing 31.** SIOT execution.

```
1    Started generation of PKA
     Started generation of PK_hat_B
     Started generation of encrypted messages c
     Started decryption of xb
5    Time elapsed (s): 11.224710386002698
```

## 5. Error-Prone Applications

The last 10 years have seen a rapid development of isogeny-based cryptography. Many new constructions and protocols are based on the SIDH key exchange from 2011. The increased interest in cryptographic systems based on isogenies was caused, among others, by similarities of SIDH to classical cryptography key exchange. New solutions are searched for in order to bring more functionalities from classical cryptography to post-quantum. Some techniques can be used: Unruh's construction, SDVS from HMAC, and PAKE using encryption. However, usage of such generic methods does not make automatically the resulting scheme secure. New construction can be vulnerable, PAKE scheme from Section 5.2 is an example. Even though the scheme is based on secure SIDH protocol and correct construction from Section 5.1 it is vulnerable. The attack [30] on that scheme is presented in Section 5.3.

All of the schemes described in Section 4, except SIDH, are based on the SIDH key exchange. The chosen schemes show specific constructions, that can turn a key exchange protocol into different types of cryptographic primitives. Some of these constructions are described in this section. Most of them are similar to constructions used in classical cryptography or present the same, generic approach.

### 5.1. Generic Constructions

Unruh's construction: As the Fiat–Shamir transform might not be secure against quantum computers, it would be very practical to have an analogous construction that is proven to be quantum-safe. The scheme described in Section 4.2 uses a quantum-resistant alternative to the Fiat–Shamir transform, known as Unruh's construction. Since the scheme is closely linked to that concept, its more detailed description is also provided in Section 4.2.

Unruh's construction transforms an interactive zero-knowledge proof system into a non-interactive one. It follows that Unruh's construction can be used to build signature schemes similarly to classical cryptography, i.e., based on previous constructions. It is a

crucial property of cryptography as it is easier to build new schemes and analyze their security if some well-studied building blocks are reused.

Unruh's construction applies to any interactive zero-knowledge proof system, not only isogeny-based ones. Thus, it is a very important building block of all post-quantum cryptography.

SDVS from HMAC: In the literature, several classical strong designated verifier signature schemes based on HMAC have been proposed, e.g., [31–33]. The general construction used in the verification algorithm of message-signature pair $(m, \sigma)$ uses the verification equation like:

$$\sigma = H(m, k), \tag{11}$$

where $H$ is some public and secure hash function, and $k$ is a key. The key $k$ can be computed as $k = y_A^{x_B} \pmod{p} = y_B^{x_A} \pmod{p}$, where values $x$ are secret keys, and $y$ are corresponding public keys of parties A and B. Any key exchange protocol can be used to compute a key for HMAC, including more advanced variants such as a pairing-based key exchange. An isogeny-based example of an SDVS scheme is described in Section 4.3.

It is sometimes argued, e.g., [34], that this construction is not a signature scheme. Most of the SDVS schemes do not have the undeniability property. Such SDVS schemes are more like a message authentication code rather than a digital signature, and that could be problematic.

PAKE using encryption: In classical cryptography, the Diffie–Hellman key exchange can be tweaked in many straightforward ways to construct PAKE schemes, e.g., EKE [35], SPEKE [36], PAK [37]. Encrypted Key Exchange (EKE) is general construction, that can add mutual authentication on top of any existing key exchange protocol. At least one party encrypts an ephemeral public key using a password, and only then sends it to a second party. The second party first decrypts the message with the same password, then continues with computation of a shared key according to the underlying protocol.

EKE-like encryption is a popular method of constructing PAKEs. This approach is used in EAP-EKE [38], and there is an isogeny-based SIDH-EKE scheme.

### 5.2. Problematic Password-Authenticated Key Establishment (PAKE) SIDH-EKE

#### 5.2.1. Construction and Algorithms

The first password-authenticated key agreement methods are Encrypted Key Exchange (EKE) methods [35]. The scheme is based on the Diffie–Hellman key exchange and introduces one modification. Instead of exchanging public keys in a clear form, the messages are encrypted with a shared password. The scheme SIDH-EKE due to Terada, and Yoneyama presented in [39] is a straightforward construction based on SIDH instead of classical DH.

All parameters of SIDH-EKE are defined as in SIDH. Let $(\mathsf{Enc}, \mathsf{Dec})$ be a symmetric key encryption scheme where $\mathsf{Enc}$ is an encryption algorithm and $\mathsf{Dec}$ is a decryption algorithm. Alice and Bob have a password *pw*.

In SIDH-EKE, public keys are generated as in SIDH but the keys are never published in a clear form. Alice and Bob both encrypt their keys with symmetric cipher and a password known only to them.

After exchanging the ciphertexts, Alice and Bob can decrypt each other's public keys with the same password. That way they achieve mutual authentication. Having public keys in a clear form, they can proceed with computations just like in the secret generation of SIDH.

#### 5.2.2. Implementation

The SIDH-EKE scheme is a simple extension of the SIDH scheme; thus, both implementations are quite similar. The implementation uses AES for public-key encryption and defines a function to transform a short password into a usable AES key.

**Comment** (Listing 32): The get_key function is a wrapper for the SHA3-256 hash function that returns 16 byte long keys for any input pw. Parties Alice and Bob are implemented as objects of a modified class Entity.

**Listing 32.** Helper function for SIDH-EKE.

```python
def get_key(pw):
    h_obj = SHA3_256.new()
    h_obj.update(f'{pw}'.encode())
    return h_obj.digest()[:16]
```

**Comment** (Listing 33): The constructor __init__ now also accepts a password pw. The method implements Algorithm 15 and generates a secret key as in SIDH. In line 17, the corresponding public key is computed in a clear form. An encryption key K is generated in line 18 and used for a new AES cipher in line 19. A Python dictionary with an encrypted key and a nonce is saved instead of a clear form key. The method gen_shared_key implements Algorithm 16. The method takes another party, i.e., peer object as a parameter. In line 27, once again the AES key is generated. If both parties use the same password pw, they will have the same encryption key. In line 29, the public key of the other party is decrypted. Starting in line 30, the SIDH secret key generation is executed.

**Listing 33.** Entity class for SIDH-EKE.

```python
lass Entity:
    def __init__(self, name, pw):
        self.name = name
        self.pw = pw
        self.P = params[name][0]
        self.Q = params[name][1]
        self.l = params[name][2]
        self.e = params[name][3]

        self.sk = random.randrange(self.l ** self.e)
        self.S = self.P + self.sk * self.Q
        # assert self.l ** self.e == self.S.order()
        self.pk = self.gen_pub_key(get_other(self.name))

    def gen_pub_key(self, other):
        pk_to_enc = isogeny_graph_walk(E, self.S, self.l, self.e, other[0], other[1])
        K = get_key(self.pw)
        cipher = AES.new(K, AES.MODE_EAX)
        return {
            'ciphertext': cipher.encrypt_and_digest(pickle.dumps(pk_to_enc)),
            'nonce': cipher.nonce
        }

    def gen_shared_key(self, peer):
        K = get_key(self.pw)
        cipher = AES.new(K, AES.MODE_EAX, nonce=peer.pk['nonce'])
        pk_dec = pickle.loads(cipher.decrypt(peer.pk['ciphertext'][0]))
        S = pk_dec[1] + self.sk * pk_dec[2]
        shared_curve, _, _ = isogeny_graph_walk(pk_dec[0], S, self.l, self.e)
        return shared_curve.j_invariant()
```

---

**Algorithm 15:** $\mathsf{KeyGen}(pw)$

| Alice | Bob |
|---|---|
| $\mathsf{sk}_A \overset{R}{\leftarrow} \mathbb{Z}/\ell_A^{e_A}\mathbb{Z}$ | $\mathsf{sk}_B \overset{R}{\leftarrow} \mathbb{Z}/\ell_B^{e_B}\mathbb{Z}$ |
| $S_A = P_A + \mathsf{sk}_A\, Q_A$ | $S_B = P_B + \mathsf{sk}_B\, Q_B$ |
| $\phi_A : E \to E_A = E/\langle S_A \rangle$ | $\phi_B : E \to E_B = E/\langle S_B \rangle$ |
| $\mathsf{pk}_A = \mathsf{Enc}_{pw}(E_A, \phi_A(P_B), \phi_A(Q_B))$ | $\mathsf{pk}_B = \mathsf{Enc}_{pw}(E_B, \phi_B(P_A), \phi_B(Q_A))$ |

**return** $(\mathsf{pk}_A, \mathsf{pk}_B)$

---

**Algorithm 16:** SessionKeyGen$(\mathsf{pk}_A, \mathsf{pk}_B)$

---

| Alice | Bob |
|---|---|
| $(E_B, \phi_B(P_A), \phi_B(Q_A)) = \mathsf{Dec}_{pw}(pk_B)$ | $(E_A, \phi_A(P_B), \phi_A(Q_B)) = \mathsf{Dec}_{pw}(pk_A)$ |
| $S'_A = \phi_B(P_A) + \mathsf{sk}_A \, \phi_B(Q_A)$ | $S'_B = \phi_A(P_B) + \mathsf{sk}_B \, \phi_A(Q_B)$ |
| $\phi'_A : E_B \rightarrow E_{AB} = E/\langle S'_A \rangle$ | $\phi'_B : E_A \rightarrow E_{BA} = E/\langle S'_B \rangle$ |
| $j_{AB} = j(E_{AB})$ | $j_{BA} = j(E_{BA})$ |

**return** $(j_{AB}, j_{BA})$

---

**Comment** (Listing 34): The above listing shows an example code that executes SIDH-EKE. First, an Entity A is created for Alice with a simple password 123456, then an Entity B for Bob with the same password. Alice can generate her shared value with Bob by passing object B as an argument to the gen_shared_key method in line 10. Bob computes his shared key with Alice in line 13 by passing object A as an argument to the gen_shared_key method. Both Alice and Bob must compute the same value, that condition is asserted in line 16.

**Listing 34.** SIDH-EKE key agreement.

```
1   t0 = time.perf_counter()

    print('Started generation of PKA')
    A = Entity('A', 123456)
5
    print('Started generation of PKB')
    B = Entity('B', 123456)

    print('Started generation of secA')
10  secA = A.gen_shared_key(B)

    print('Started generation of secB')
    secB = B.gen_shared_key(A)

15  t1 = time.perf_counter()
    assert secA == secB
    print("Time elapsed (s):", t1 - t0)
```

**Comment** (Listing 35): The output of the above program is presented in this listing, it shows that the assertion succeeded.

**Listing 35.** SIDH-EKE execution.

```
1   Started generation of PKA
    Started generation of PKB
    Started generation of secA
    Started generation of secB
5   Time elapsed (s): 8.174498736999794
```

### 5.3. Attack on SIDH-Eke

Although EKE scheme is secure in a classical setting and EAP-EKE is successfully used in practice, the isogeny-based straightforward construction SIDH-EKE from Section 5.2 is susceptible to man-in-the-middle and offline dictionary attacks. The following attack was presented in [30].

For the SIDH-EKE scheme to be secure, public keys must be indistinguishable from random bitstrings. This property is necessary to fulfill the offline dictionary attack resistance security requirement of PAKE. However, in practice, public keys of SIDH are distinguishable from random bitstrings; thus, it is possible to construct an oracle that can determine if a password guess was correct.

A passive eavesdropper Eve can observe SIDH-EKE execution between Alice and Bob. After key generation Algorithm 15 finishes, Alice and Bob exchange encrypted public keys $\mathsf{pk}_A = \mathsf{Enc}_{pw}(E_A, \phi_A(P_B), \phi_A(Q_B))$, and $\mathsf{pk}_B = \mathsf{Enc}_{pw}(E_B, \phi_B(P_A), \phi_B(Q_A))$. Eve can perform an offline dictionary attack on Alice's public key as shown in Algorithm 17, analogous approach can be applied for public key of Bob.

In SIDH, a public key is a structured data set, a triple of the form $(E, \phi(P), \phi(Q))$, where $E$ is a supersingular elliptic curve, and $\phi(P), \phi(Q)$ are basis points. The authors of SIDH-EKE claim the scheme prevents offline dictionary attacks because the attacker cannot determine if a password guess is valid or not. That is because the encryption scheme is modeled as an ideal cipher. However, it is simple to check if the decryption of a public key yields valid values. Eve can observe Alice sending her public key $\mathsf{pk}_A$, and then try to guess a password $pw'$. For each password, Eve decrypts the key $\mathsf{Dec}_{pw'}(\mathsf{pk}_A)$, parses the values $(E'_A, \phi_A(P_B)', \phi_A(Q_B)')$ and checks all properties according to Algorithm 17. For a random password, the probability that even some of these criteria are met is extremely low, so if all checks succeed, $pw'$ is a correct guess with high probability.

---

**Algorithm 17:** SIDH offline dictionary attack

---

> observe $\mathsf{pk}_A$
> guess $pw'$ and decrypt $(E'_A, \phi_A(P_B)', \phi_A(Q_B)') = \mathsf{Dec}_{pw'}(\mathsf{pk}_A)$
> **check** $E'_A$ is supersingular
> **check** $\phi_A(P_B)', \phi_A(Q_B)'$ lie on $E'_A$
> **check** $\phi_A(P_B)', \phi_A(Q_B)'$ have order $\ell_B^{e_B}$
> **check** the Weil pairing $e(\phi_A(P_B)', \phi_A(Q_B)')$ is maximal
> **if** *all checks succeed* **then**
> $\quad$ | $\quad$ **return** $pw'$
> **end**

---

The same reasoning applies to optimized practical implementations of SIDH and SIKE. Even when the public parameters are compressed, e.g., sending the $x$-coordinates $\phi_A(P_B)$, $\phi_A(Q_B)$, and $\phi_A(Q_B - P_B)$ instead of directly sending the elliptic curve, enough information is sent to recover $E_A, \phi_A(P_B)$, and $\phi_A(Q_B)$. Thus, the offline dictionary attack is still applicable.

The original EKE scheme is based on discrete logarithm; thus, unencrypted public keys are just very large numbers. A decryption of a public key with a random password would produce a bitstring that can be interpreted as a number, so it could be a valid public key. As a result, a password oracle similar to SIDH-EKE does not exist. An elliptic curve discrete logarithm variant of EKE would be also vulnerable to offline dictionary attacks.

Another isogeny-based PAKE is presented in [40]. The protocol does not use encryption of public keys, instead, auxiliary points are obfuscated with a reversible algebraic operation.

**Remark 3.** *Some of isogeny-based schemes use exactly the same or very similar, generic constructions known in classical cryptography to achieve new security properties from other types of cryptographic primitives. Analogies between SIDH and DH may suggest that some constructions should be secure against attacks against quantum computers. However, cryptography is tricky and it is not always true. Some mistakes may render isogeny-based schemes vulnerable even against classical computers. SIDH-EKE is an example and a valuable lesson.*

## 6. Computational Complexity and Benchmarks

This section contains time measurements of the implementations. The previous sections show only executions of the scheme as a whole together with time measurements from one run. Averaged time measurement in seconds of many runs of each of the algorithms from Section 4 is also provided. When applicable, different schemes are also compared. Tests were run on a single core of Intel Core i7-9750H processor.

### 6.1. Supersingular Isogeny Diffie–Hellman Key Exchange (SIDH)

Table 2 contains measurements of running times of the key generation and secret generation algorithms. The benchmark was run using SIKEp434 [16] parameters, i.e., $p = 2^{216}3^{137} - 1$.

**Table 2.** Time measurements of SIDH in seconds.

|  | Key Generation | Secret Generation |
|---|---|---|
| Alice | 2.43 | 2.11 |
| Bob | 2.02 | 1.74 |

From Table 2 it is possible to draw two conclusions.

- Bob can generate his key and a shared secret faster than Alice. It follows from the choice of parameters, Alice has to compute 216 isogenies of degree 2, while Bob computes 137 isogenies of degree 3. For different parameters from SIKE specification, Bob always needs to compute a smaller number of isogenies than Alice.
- Although Alice always computes 216 isogenies of degree 2, and Bob computes 137 isogenies of degree 3, for both parties secret value generation is faster than a key generation. The difference between the key and shared value generation algorithms is that during secret value generation no auxiliary points need to be moved, thus allowing shorter running times.

### 6.2. Isogeny-Based Digital Signature (Ibds)

Table 3 contains measurements of running times of the key generation, signing, and verification algorithms. The benchmark was run using SIKEp434 [16] parameters, i.e., $p = 2^{216}3^{137} - 1$, and the security parameter $\lambda = 8$.

**Table 3.** Time measurements of IBDS in seconds.

| Key Generation | Signing | Verification |
|---|---|---|
| 2.40 | 63.46 | 44.48 |

Table 3 shows results matching Table 2. The running time of the IBDS key generation algorithm is similar to the running time of key generation of Alice in SIDH. It is expected since those algorithms perform the same computations. The running time of the signing algorithm also matches the expected time, i.e., it is about $2\lambda$ times longer than SIDH Bob's key and secret generation. The running time of the verification algorithm matches the expected time of $2\lambda \cdot 1.5$ times longer than SIDH Bob's secret generation. The 1.5 cofactor is a result of the fact that depending on the challenge bit, 2 or 1 values of the response are validated in each of the $2\lambda$ rounds.

### 6.3. Strong Designated Verifier Signature (SDVS)

Table 4 contains measurements of running times of the key generation, signing, and verification algorithms. The benchmark was run using SIKEp434 [16] parameters, i.e., $p = 2^{216}3^{137} - 1$.

**Table 4.** Time measurements of SDVS in seconds.

|  | Key Generation | Signing | Verification |
|---|---|---|---|
| Signer | 2.45 | 2.02 | - |
| Verifier | 1.97 | - | 1.72 |

The SDVS scheme is heavily based on the SIDH key exchange. The running times of the key generation algorithm for a signer and a verifier are similar to the times of SIDH Alice and Bob, respectively. The time needed for signing matches the time of Alice's secret generation, while verification takes as much as Bob's secret generation. All of these results are expected.

### 6.4. Undeniable Signatures

Table 5 contains measurements of running times of the key generation, signing, confirmation, and disavowal algorithms. The benchmark was run using toy-size parameters, i.e., $p = 3^3 2^4 5^2 - 1$, and the security parameter $\lambda = 128$.

**Table 5.** Time measurements of undeniable signatures in seconds.

| Key Generation | Signing | Confirmation | Disavowal |
|:---:|:---:|:---:|:---:|
| 0.0058 | 0.014 | 7.93 | 6.22 |

In the case of the undeniable signatures implementation, it is difficult to compare the results to the previous implementations. The scheme differs the most from the other schemes and the implementation uses a different set of parameters. Comparing orders of magnitude of running times of the key generation algorithms across all implementations it is expected that running times of the confirmation and disavowal protocols will increase at least a few hundred times if parameters matching SIKEp434 are used. The difference between times of the confirmation and disavowal protocols in Table 5 is caused by the fact that in each round of the confirmation protocol a signer checks the validity of a signature before a commitment is returned. The time needed for the confirmation protocol can be reduced to time close to the running time of disavowal protocol if each of the $\lambda$ rounds is not treated as a completely independent execution, i.e., a signer would check a signature only once and then return $\lambda$ commitments.

### 6.5. Supersingular Isogeny Oblivious Transfer (SIOT)

Table 6 presents running times of algorithms executed in SIOT. The benchmark was run using SIKEp434 [16] parameters, i.e., $p = 2^{216} 3^{137} - 1$.

**Table 6.** Time measurements of SIOT in seconds.

|  | Key Generation | Encryption | Decryption |
|:---:|:---:|:---:|:---:|
| Alice | 2.36 | 4.17 | — |
| Bob | 1.93 | — | 1.76 |

Times measured during the key generation algorithm execution are very similar to the times of corresponding values in Table 2. Both parties in SIOT need to execute additional steps on top of SIDH computations. Alice chooses two random messages while Bob chooses bit $b$ and performs some point computations on an elliptic curve. However, all of those new steps are inexpensive when compared to computational cost of isogeny computations.

The time needed for decryption of the message chosen by Bob matches the time of Bob's secret generation in SIDH. It is expected since the only additional step in SIOT is the decryption of the message with the shared secret as the key. Alice needs about twice as much time for encryption as Alice needs to compute shared secret in SIDH. That correlation follows from the fact that in SIOT Alice has to compute two different keys-one for each of the messages.

### 7. Conclusions

The goal of this article was a review of the state-of-the-art functionalities provided by existing isogeny-based cryptosystems. The schemes include key exchange protocol, oblivious transfer, and several types of digital signatures. For each of the schemes, similarities between them and analogies to classical cryptography were highlighted. The research covers a detailed description of chosen isogeny-based schemes. Software solutions for isogeny-based cryptography were briefly discussed and the chosen schemes were implemented. The goal of implementations was not maximum performance but clarity of code that could help to understand the schemes and isogeny-based cryptography in

general. Common blocks of the implementations can be used for fast prototyping of new cryptosystems. In the end, time benchmarks were provided for comparison.

From the conducted research and implementation work, the following conclusions can be drawn.

- The properties of SIDH imply it can be a natural candidate to replace commonly used Diffie–Hellman and elliptic curve Diffie–Hellman key exchange.
- Analogies between SIDH and DH caused rapid development of isogeny-based cryptosystems using constructions analogous to classical cryptography.
- SIDH key can be used for symmetric encryption schemes, similarities of SIDH and DH allow building ElGamal-like public key encryption.
- Isogeny-based cryptography needs a more cautious approach, PAKE that is badly designed in an obvious way is a valuable lesson.
- Most of the post-quantum isogeny signatures are far from being practical; however, some more practical signatures exist, e.g., CSI-FiSh [41].

**Author Contributions:** Conceptualization, Ł.K.; data curation, B.D. and Ł.K.; resources, Ł.K.; software, B.D.; writing—original draft, B.D.; writing—review and editing, B.D. and Ł.K. All authors have read and agreed to the published version of the manuscript.

**Funding:** The research was partially financed from the Fundamental Research Fund number 8211104160 of the Wrocław University of Science and Technology.

**Data Availability Statement:** Not applicable.

**Conflicts of Interest:** The authors declare no conflict of interest.

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
