# Peer review of "Review of Chosen Isogeny-Based Cryptographic Schemes"

_cryptography, doi:10.3390/cryptography6020027_

Round 1

Reviewer 1 Report

I think this is an interesting survey of one of the approaches to post-quantum cyptography, and merits publication. My request would be that a brief section can be introduced to explain to the reader familiar with quantum information processing why the chosen isogeny-based schemes are quantum-resistant. 

Author Response

Thank you for your suggestion. A section about quantum resistance is indeed required. In the revised manuscript, we have included a few paragraphs that discuss possible classical attacks and a hypothesis about why quantum computers cannot improve classical attacks.

Reviewer 2 Report

The article is very interesting and relevant. Although the rate of progress of quantum computers is difficult to predict, the development and analysis of new, quantum computer-resistant, open-key cryptographic algorithms and protocols are critical. This is illustrated by the competition published by NIST for standards for postquantum algorithms.

The article is written very clearly and consistently and is easy to read. However, the question of why the works submitted for review were chosen is not sufficiently substantiated. The paper also presents a specific implementation of the algorithms but does not indicate how new they are, namely the results obtained by the authors

Author Response

Thank you for your suggestions. In the revised manuscript, we have included
1. A short justification of the choice of protocols to review.
2. A clarification about the goal of the implementations. We note that our goal was not the speed improvement of computations. We want to use implementations written in Python/SAGE to help the reader to understand the chosen cryptosystems. As far as we know, there is no publication that tries to present isogeny-based cryptosystems in such a way. Some isogeny publications do include implementations but the focus is on maximal performance. The code is not that easy to understand, and sometimes it is even hard to compile and test the provided software. Our implementation is intended to be easy to comprehend, modify and run.

Reviewer 3 Report

A survey is given of fundamental isogeny-based schemes with emphasis on practical aspects. For sure the exposed programs in SAGE make easier to analyze isogeny-based cryptography.

The article is quite original since it provides a concise exposition of the mathematical methods and together with the provided program codes it is possible for any reader to gain practice in concrete and effective calculations. This approach is not common on research articles, hence it is rather correct to point out by the authors the character of survey type for the reviewed paper. 

Cryptography based on isogenies has already a long history and most probably it is a firm candidate to be designated as the standard for post-quantum cryptography. But the technical literature is focused on the mathematical methods and on reports of implementations, explaining limited experiences of some groups, in general without providing access to the developed programs. Instead, in the reviewed paper with just copy-and-paste of the programming codes any reader may test by himself the exposed procedures. 

The surveyed procedures and the chosen parameters for the domain elliptic curves are those used in the current tested implementations to be included in the forthcoming standards. 

Thus I strongly recommend the acceptation of the reviewed paper. This survey would help not just students attending courses following well known booktexts on modern cryptography but as well as working cryptographers involved on implementation and development of cryptographic schemes. 

Author Response

Thank you for your review.
If we are not mistaken, there are no suggestions for corrections in the review. However, if we did miss some points please let us know.